# Epigenomic landscapes of retinal rods and cones

Alisa Mo[1,2], Chongyuan Luo[3,4], Fred P Davis[5†], Eran A Mukamel[6], Gilbert L Henry[5], Joseph R Nery[3], Mark A Urich[3], Serge Picard[5], Ryan Lister[3,7], Sean R Eddy[5‡§], Michael A Beer[8,9], Joseph R Ecker[3,4], Jeremy Nathans[1,2,10,11]*

[1]Department of Molecular Biology and Genetics, Johns Hopkins University School of Medicine, Baltimore, United States; [2]Department of Neuroscience, Johns Hopkins University School of Medicine, Baltimore, United States; [3]Genomic Analysis Laboratory, The Salk Institute for Biological Studies, La Jolla, United States; [4]Howard Hughes Medical Institute, The Salk Institute for Biological Studies, La Jolla, United States; [5]Janelia Research Campus, Howard Hughes Medical Institute, Ashburn, United States; [6]Department of Cognitive Science, University of California San Diego, La Jolla, United States; [7]The ARC Centre of Excellence in Plant Energy Biology, The University of Western Australia, Crawley, Australia; [8]McKusick-Nathans Institute of Genetic Medicine, Johns Hopkins University School of Medicine, Baltimore, United States; [9]Department of Biomedical Engineering, Johns Hopkins University, Baltimore, United States; [10]Department of Ophthalmology, Johns Hopkins University School of Medicine, Baltimore, United States; [11]Howard Hughes Medical Institute, Johns Hopkins University School of Medicine, Baltimore, United States

*For correspondence: jnathans@ jhmi.edu

Present address: †Molecular Immunology and Inflammation Branch, National Institute of Arthritis and Musculoskeletal and Skin Diseases, National Institutes of Health, Bethesda, United States; ‡Howard Hughes Medical Institute, Harvard University, Cambridge, United States; §Department of Molecular & Cellular Biology and John A. Paulson School of Engineering and Applied Sciences, Harvard University, Cambridge, United States

**Abstract** Rod and cone photoreceptors are highly similar in many respects but they have important functional and molecular differences. Here, we investigate genome-wide patterns of DNA methylation and chromatin accessibility in mouse rods and cones and correlate differences in these features with gene expression, histone marks, transcription factor binding, and DNA sequence motifs. Loss of NR2E3 in rods shifts their epigenomes to a more cone-like state. The data further reveal wide differences in DNA methylation between retinal photoreceptors and brain neurons. Surprisingly, we also find a substantial fraction of DNA hypo-methylated regions in adult rods that are not in active chromatin. Many of these regions exhibit hallmarks of regulatory regions that were active earlier in neuronal development, suggesting that these regions could remain undermethylated due to the highly compact chromatin in mature rods. This work defines the epigenomic landscapes of rods and cones, revealing features relevant to photoreceptor development and function.

## Introduction

The retina is the starting point of vision. It originates from the embryonic diencephalon and contains three layers of neurons: an outer nuclear layer with rods and cones; an inner nuclear layer with bipolar, horizontal, and amacrine cells; and a ganglion cell layer (*Swaroop et al., 2010*). Rods can respond to a single photon and mediate vision in dim light. Cones are less sensitive to light and mediate color vision. Photoreceptor specialization results from well-defined rod- and cone-specific patterns of gene expression (*Kefalov, 2012*; *Siegert et al., 2012*), which are in part controlled by retinal transcription factors (TFs) OTX2, CRX, NRL, and NR2E3. In both rods and cones, OTX2

**eLife digest** Vision in humans is made possible by a light-sensing sheet of cells at the back of the eye called the retina. The surface of the retina is populated by specialized sensory cells, known as rods and cones. The rod cells detect very dim light, while the cones are less sensitive to light but are used to detect color. Together, the rods and cones gather the information needed to create a picture that is then transmitted to the brain.

Rods and cones have been studied for decades, and genetic analyses have revealed the patterns of gene expression that lead a cell to develop into either a rod or a cone. Researchers have also identified several key regulatory genes that control these patterns, but less is known about the role of other factors that control the expression of genes.

Chemical modifications to DNA or modifications to the proteins associated with DNA – which are collectively called epigenetic modifications – can either promote or inhibit the activation of nearby genes. Now, Mo et al. have shown that rods and cones from mice have very different patterns of epigenetic modifications. The experiments also revealed that many sections of DNA that are marked to promote gene activation contain known rod-specific or cone-specific genes; and that rod cells need a known regulatory gene to develop their specific pattern of epigenetic modifications. Finally, Mo et al. showed that epigenetic regulation differed between brain cells and rods and cones.

These insights into epigenetic regulation of rod and cone genes may help explain why some people with eye diseases caused by the same genetic mutation may develop symptoms at different ages or lose vision at different rates. The new information about gene regulation may also help scientists to reprogram stem cells to become healthy rods or cones that could be transplanted into people with eye disease to restore their vision.

determines photoreceptor cell fate (*Nishida et al., 2003*), and CRX regulates expression of terminal photoreceptor genes (*Furukawa et al., 1997*). Rod photoreceptor fate and gene activation are induced by NRL and its downstream target NR2E3 (*Mears et al., 2001*). Loss of NR2E3 leads to enhanced S-cone syndrome, an autosomal recessive human retinal disease (*Haider et al., 2000*) that is recapitulated in retinal degeneration 7 (*rd7*) mice (*Haider et al., 2001*). *rd7* rods show a partial conversion of photoreceptor identity because they retain expression of rod-specific genes but also de-repress a subset of cone-specific genes (*Chen et al., 2005*; *Corbo and Cepko, 2005*; *Peng et al., 2005*).

Regulatory regions such as enhancers and promoters control functional differences between rods and cones. Although these regions are beginning to be defined, current studies have limitations. ChIP-seq can identify TF binding sites but requires high-quality antibodies and can only interrogate one TF at a time. TF binding sites are typically marked by increased chromatin accessibility (*Thurman et al., 2012*). However, existing datasets measuring chromatin accessibility are limited to whole retina from wild-type mice (*Wilken et al., 2015*). Because rods make up 70–80% of all mouse retinal cells and outnumber cones by 35:1 (*Jeon et al., 1998*), whole retina studies can approximate features of rods but mask differences between rods and cones that contribute to their unique identities. Therefore, the current understanding of photoreceptor gene regulation also remains limited by the lack of cell type-specific information.

Of special interest is the high positive correlation between accessible chromatin and local regions of low DNA methylation that has been observed in various cell types (*Stadler et al., 2011*; *Hon et al., 2013*; *Ziller et al., 2013*; *Mo et al., 2015*). TF binding can result in local regions of low DNA methylation, leading to strong overlaps between regions identified as DNA hypo-methylated and as accessible chromatin. At present, genome-wide, single-base resolution DNA methylation profiles have not been reported for either rods or cones, precluding a large-scale analysis of this phenomenon in either photoreceptor type. Also of interest is the small size of the rod nucleus (~5 µm; *Solovei et al., 2009*) and its highly condensed chromatin (*Kizilyaprak et al., 2010*), which may potentially impact how chromatin accessibility correlates with DNA methylation. In addition, rods are the only known cell type in mice with nuclei that have heterochromatin centers surrounded by peripheral euchromatin (*Carter-Dawson and LaVail, 1979*). This inverted organization is thought to

facilitate nocturnal vision (*Solovei et al., 2009*). By contrast, cone nuclei are larger and exhibit the conventional arrangement of central euchromatin and peripheral heterochromatin.

Here, we explore the epigenomic differences that contribute to rod and cone photoreceptor identity. Unexpectedly, most rod-specific regions of low DNA methylation are not located in accessible chromatin in adult rods. Instead, our evidence suggests that these regions are potential active regulatory sites in fetal neural tissue and, despite loss of active chromatin marks, remain hypo-methylated in adult rods due to the barrier to cytosine methyltransferases posed by chromatin condensation. In addition, we identify rod- and cone-enriched regions of accessible chromatin that may play gene regulatory functions and carry distinct DNA sequence motifs. Integrated analysis of *rd7* rods, together with normal rods and cones, shows that NR2E3 function is necessary for rods to gain their complete ensemble of epigenomic features. We further examine epigenomic patterns in retinal photoreceptors versus brain neurons. Overall, our findings highlight both global and local epigenomic differences between retinal rods and cones that reflect unique aspects of their biology.

## Results

### Compared to cones, rods have a larger fraction of hypo-methylated DNA that is discordant with accessible chromatin

To characterize putative regulatory DNA in adult rod and cone photoreceptors, we purified their nuclei using either affinity purification (INTACT; *Mo et al., 2015*) or flow cytometry (*Figure 1—figure supplement 1*). We then applied ATAC-seq to map sites of enhanced chromatin accessibility that include putative sites of TF binding (*Buenrostro et al., 2013*), and we applied MethylC-seq to measure DNA methylation levels at single-base resolution (*Lister et al., 2008*) (*Figure 1A*; *Supplementary file 1*). All samples were analyzed using independent pairs of biological replicates.

We first assessed the genome-wide relationship between DNA methylation and chromatin accessibility in rod and cone photoreceptors. Using previously defined criteria (*Stadler et al., 2011*; *Burger et al., 2013*), we identified two types of regions that are depleted for DNA methylation: (1) 16617 rod and 15888 cone discrete (<5 kb) un-methylated regions (UMRs; median mCG = 6%), which tend to be at promoters, and (2) 85992 rod and 65791 cone low-methylated regions (LMRs; median mCG = 24%), which are likely associated with distal regulatory regions (*Supplementary file 2*). We identified a set of 55366 regions in rods and 75650 regions in cones with increased ATAC-seq densities that mark accessible chromatin (*Supplementary file 3*).

In both rods and cones, ATAC-seq peaks exhibit low levels of DNA methylation (*Figure 1B*). On average, hypo-methylated regions also have elevated ATAC-seq signals (*Figure 1C*), and UMRs show strong overlap with ATAC-seq peaks (*Figure 1D*). However, we were surprised to find substantial differences between rods and cones in the fraction of LMRs that are located within accessible chromatin (*Figure 1D*). 38% (24830) of cone LMRs overlap with ATAC-seq peaks, a percentage close to previous reports in embryonic stem cells and cortical neurons (*Xie et al., 2013*; *Mo et al., 2015*). Yet, only 19% (15984) of rod LMRs overlap with ATAC-seq peaks. In fact, 31% of rod LMRs show no sign of chromatin accessibility (<0.1 ATAC-seq FPKM), whereas this fraction is only 7% for cones. This analysis shows that rods harbor a substantial compartment of demethylated, but inaccessible, DNA that is largely absent in cones.

To further explore this observation, we applied ATAC-seq to an independent set of retinal samples that did not require nuclear purification. Approximately 70–80% of cells in the WT mouse retina are rods (*Jeon et al., 1998*) and, in the absence of NRL, cells fated to become rods are converted *en masse* to S-cones (*Mears et al., 2001*). Therefore, the sites of accessible chromatin in unfractionated nuclei from WT and *Nrl KO* retinas would be expected to largely mirror those in rods and cones, respectively. Similar to our results using purified nuclei, a greater fraction of cone LMRs overlap *Nrl KO* ATAC-seq peaks (49%; 31952) compared to rod LMRs that overlap WT ATAC-seq peaks (26%; 22590) (*Figure 1—figure supplement 2*).

### Rod-specific hypo-methylated regions are putative active regulatory regions in fetal neural tissue

A previous study demonstrated that a subset of active enhancers in embryonic tissue have low levels of DNA methylation in adult tissues in the absence of ongoing enhancer activity (*Hon et al., 2013*).

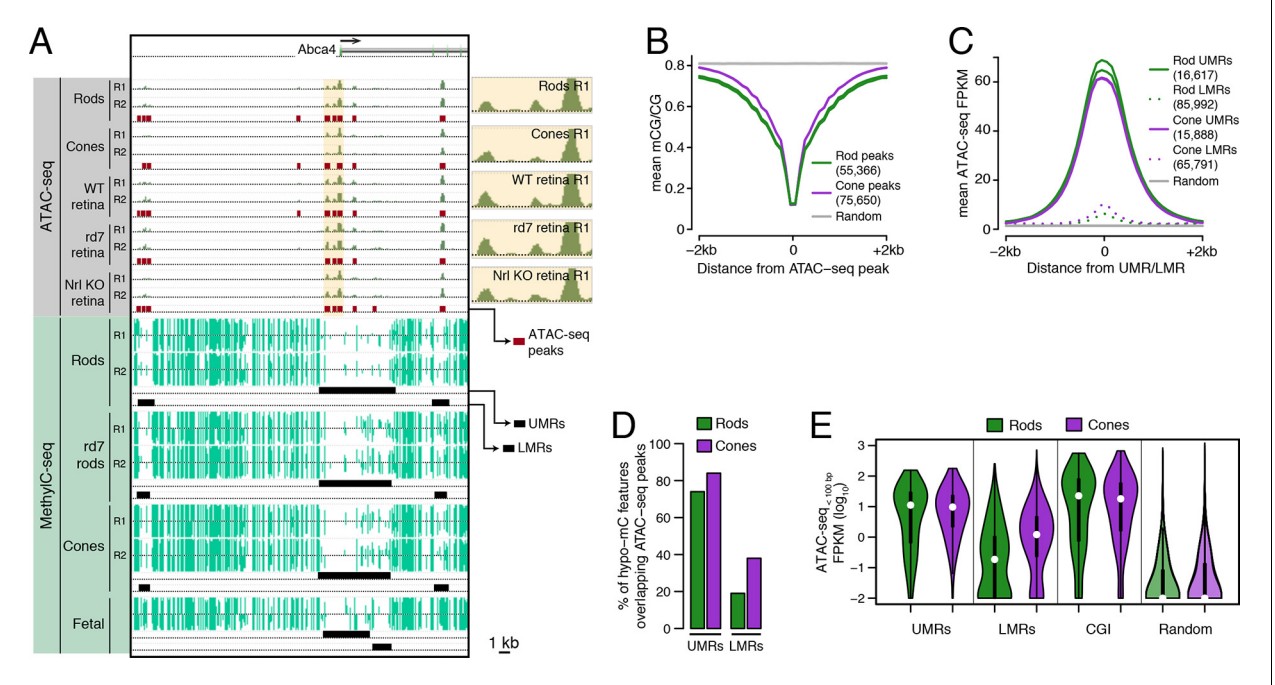

**Figure 1.** Relationship of DNA methylation and accessible chromatin in retinal rods and cones. (**A**) Browser image showing accessible chromatin (top) and DNA methylation (bottom) near *Abca4*, a photoreceptor gene expressed by both rods and cones. Enlarged images of ATAC-seq signals in the highlighted area are shown for one replicate of each cell or tissue type. For ATAC-seq, <100 bp ATAC-seq reads are shown. For DNA methylation, mCG/CG is shown. Methylated CG positions are indicated by upward (plus strand) and downward (minus strand) ticks, with the height of each tick representing the fraction of methylation at the site ranging from 0 to 1. Bars below the raw data show locations identified as ATAC-seq peaks, UMRs, and LMRs. Fetal, fetal E13 cerebral cortex from *Lister et al., 2013*. Biological replicates (R1, R2). (**B**) Line plot showing lower mean CG methylation at rod and cone ATAC-seq peaks relative to size-matched random genomic regions (repeated 10 times). (**C**) Line plot showing higher mean ATAC-seq signals at rod and cone UMRs and LMRs relative to size-matched random genomic regions (repeated 10 times). (**D**) Barplot showing that the percentage of cone LMRs that overlap ATAC-seq peaks (38%) is two-fold higher than the percentage of rod LMRs that overlap ATAC-seq peaks (19%). (**E**) Violin plot showing that rod LMRs have a bimodal distribution of <100 bp ATAC-seq signals. The median (white dot) and interquartile range (black bar) are indicated. CGI: CpG islands; Random: size-matched random genomic regions (repeated 10 times).

The following figure supplements are available for figure 1:

**Figure supplement 1.** Genetic labeling of mouse rod and cone photoreceptor nuclei.

**Figure supplement 2.** Accessible chromatin in whole retina versus DNA methylation.

Referred to as vestigial enhancers, these hypo-methylated regions are not enriched for active histone marks and DNaseI hypersensitivity in adult cells. When we examine the densities of sub-nucleosomal-length ATAC-seq reads, a subset of reads that may better capture sites of TF binding (*He et al., 2014*), we observe a bimodal distribution of rod ATAC-seq signals at LMRs (*Figure 1E*), with the lower peak at nearly zero signal. This distribution is significantly different than that of cone ATAC-seq signals at LMRs (bootstrap Kolmogorov-Smirnov $p < 2.2 \times 10^{-16}$) and potentially reflects a greater number of vestigial enhancers in rod, compared to cone, LMRs. If this hypothesis were correct, we would expect these regions to show differential methylation between rods and cones, with rods retaining low levels of DNA methylation and cones gaining methylation. We would further expect these regions to be enriched for epigenomic marks associated with active regulatory regions in neural progenitor tissue.

To explore this idea, we first identified differentially methylated regions (DMRs) between rods and cones (*Feng et al., 2014*) (*Figure 2A*; *Supplementary file 4*). We find a greater number of DMRs that have lower methylation levels (hypo-DMRs) in rods than in cones (10784 rod hypo-DMRs versus 6693 cone hypo-DMRs). As expected from previous studies (*Schultz et al., 2015*), DNA

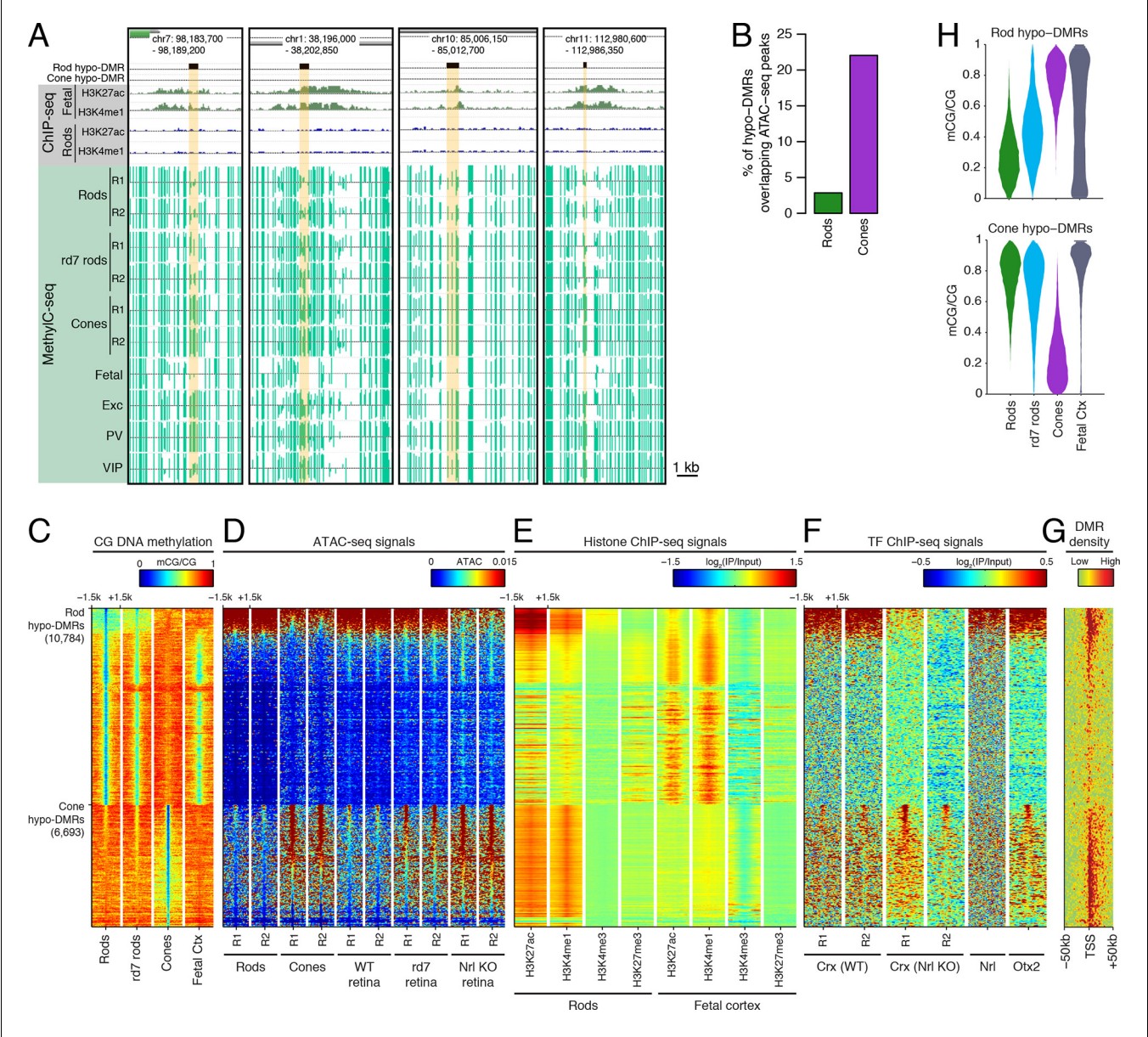

**Figure 2.** Rod hypo-DMRs show active chromatin marks in early neural development. (A) Browser images showing examples of rod hypo-DMRs that are enriched for active enhancer histone marks in fetal E14.5 brain (from *Shen et al., 2012*) but not in adult rods. These rod hypo-DMRs also display low levels of DNA methylation in both WT rods and fetal E13 cerebral cortex (from *Lister et al., 2013*), but not in cones or in most adult cortical neuron types (Exc, PV, VIP; from *Mo et al., 2015*). In addition, *rd7* rods show higher levels of methylation than WT rods but lower levels than cones. (B) Cone hypo-DMRs show a six-fold higher overlap with ATAC-seq peaks, compared to rod hypo-DMRs with rod ATAC-seq peaks. (C–G) Heatmap showing CG methylation levels in a 3 kb window centered at rod and cone hypo-DMRs (C). At the same genomic regions, the following were plotted: ATAC-seq signal (D), ChIP-seq signals for histone modifications in adult rods (this study) or in E14.5 fetal brain (from *Shen et al., 2012*) (E), and ChIP-seq signals for retinal TFs (from *Corbo et al., 2010*; *Hao et al., 2012*; *Samuel et al., 2014*) (F). The density of DMRs relative to their closest TSS is shown in (G) for a 100 kb window around the TSS. For (C–G), the rows are ordered by decreasing rank of the absolute signals of rod and cone ATAC-seq data at rod and cone hypo-DMRs, respectively. (H) The fetal cortex shares low CG DNA methylation with rods at a substantial fraction of rod hypo-DMRs (top), but shows high methylation at the majority of cone hypo-DMRs (bottom). Furthermore, methylation levels in *rd7* rods are generally intermediate between WT rods and cones, particularly at rod hypo-DMRs.

The following figure supplements are available for figure 2:

**Figure supplement 1.** Rod versus cone DNA methylation levels at DMRs are strongly anti-correlated with relative gene expression.

**Figure supplement 2.** Relationship of rod and cone hypo-DMRs to gene promoters.

methylation levels at DMRs around gene transcription start sites (TSSs) show a strong negative correlation with gene expression, with a trough Pearson correlation of -0.8 4 kb downstream of the TSS (*Figure 2—figure supplement 1*).

Similar to our results using LMRs, we find that rod hypo-DMRs are especially discordant with accessible chromatin: only 307 (3%) of rod hypo-DMRs overlap rod ATAC-seq peaks, compared with 1475 (22%) of cone hypo-DMRs that overlap cone ATAC-seq peaks (*Figure 2B*). In rods the large majority of hypo-DMRs are depleted for ATAC-seq signal (*Figures 2C–D*), active histone modifications H3K4me1 and H3K27ac (*Figure 2E*), and retinal photoreceptor TF binding (*Figure 2F*), and they are located distally from promoters (*Figure 2G*; *Figure 2—figure supplement 2*). In contrast, the minority of rod hypo-DMRs that show strong ATAC-seq signals are located at relatively closer distances to the TSS (*Figure 2G*).

We next evaluated the levels of DNA methylation (*Lister et al., 2013*) and histone modifications (*Shen et al., 2012*) in fetal cerebral cortex and brain, respectively, which are rich sources of neural progenitors and immature neurons. Supporting the idea that rod hypo-DMRs may be enriched for enhancers that function earlier in neural development, most rod hypo-DMRs have low levels of DNA methylation in fetal cortex (*Figure 2C,H*) and are enriched for active histone modifications in fetal brain (*Figure 2E*).

Could the greater discrepancy between accessible chromatin and DNA hypo-methylation in rods compared to cones be a result of greater chromatin compaction in rods? If chromatin compaction were to limit the access of cytosine methyltransferases to DNA, vestigial enhancers could remain undermethylated in adult rods. To address this question, we took advantage of the observation that disruption of NR2E3 in *rd7* rods preserves the inverted chromatin arrangement, but reduces chromatin condensation (*Corbo and Cepko, 2005*). Therefore, *rd7* rods provide a natural model to explore the relationship between chromatin condensation and the global DNA methylation pattern. We find that whereas *rd7* rods are also hypo-methylated at rod hypo-DMRs, including those that may encompass vestigial enhancers, they show higher methylation levels than WT rods (*Figure 2C,H*). Taken together, these data are consistent with a role for chromatin condensation in limiting the methylation of vestigial enhancers.

## Retinal photoreceptors show cell type-specific and shared features of accessible chromatin

To identify putative regulatory regions in rod and cone photoreceptors, we analyzed their patterns of chromatin accessibility. We first evaluated the cellular specificity of accessible chromatin, reasoning that differences in accessibility would help pinpoint regulatory regions important for unique aspects of rod and cone identity. For this analysis, we focused on comparisons between WT and *Nrl KO* retinas, rather than between purified rod and cone nuclei, because the complete absence of rods in the *Nrl KO* retina provides a degree of purity that cannot be obtained by physical separation. We note, however, that the data obtained from purified rod and cone nuclei closely match those obtained from WT and *Nrl KO* retinas (*Figure 3—figure supplement 1*). Furthermore, ATAC-seq signals between biological replicates are highly similar at accessible chromatin (Pearson r >0.99; *Figure 3—figure supplement 2*).

In comparing WT and *Nrl KO* retinas, 22520 ATAC-seq peaks show >2-fold increased accessibility in *Nrl KO* retina, but only about a third as many (7916 peaks) have greater accessibility in WT retina (*Figure 3A*). WT-enriched regions of accessible chromatin cluster near the promoters of rod-specific genes (*Supplementary file 5*) and have high levels of H3K27ac, H3K4me1, and H3K4me3 (*Figure 3B–D*). In contrast, *Nrl KO*-enriched accessible chromatin cluster near promoters of cone-specific genes (*Figure 3B,C*). We further examined previously published ChIP-seq data for OTX2 (*Samuel et al., 2014*), NRL (*Hao et al., 2012*), and CRX (from both WT and *Nrl KO* retina; *Corbo et al., 2010*). A higher percentage of binding sites for NRL and CRX in WT retina overlap WT-enriched, rather than *Nrl KO*-enriched, peaks (*Figure 3E*). However, CRX binding sites in *Nrl KO* retina show higher overlap with *Nrl KO*-enriched peaks.

WT and *Nrl KO* retina also share regions of common accessible chromatin, including 43104 accessible chromatin regions with <2-fold difference in ATAC-seq signal (*Figure 3A*). Furthermore, the WT retina shows lower, but non-zero, levels of accessibility compared to the *Nrl KO* retina at many ATAC-seq peaks near cone-specific genes (*Figure 3C*). The analogous result is seen at rod-specific genes, with lower-amplitude *Nrl KO* retina ATAC-seq signals at sites of higher-amplitude WT retina

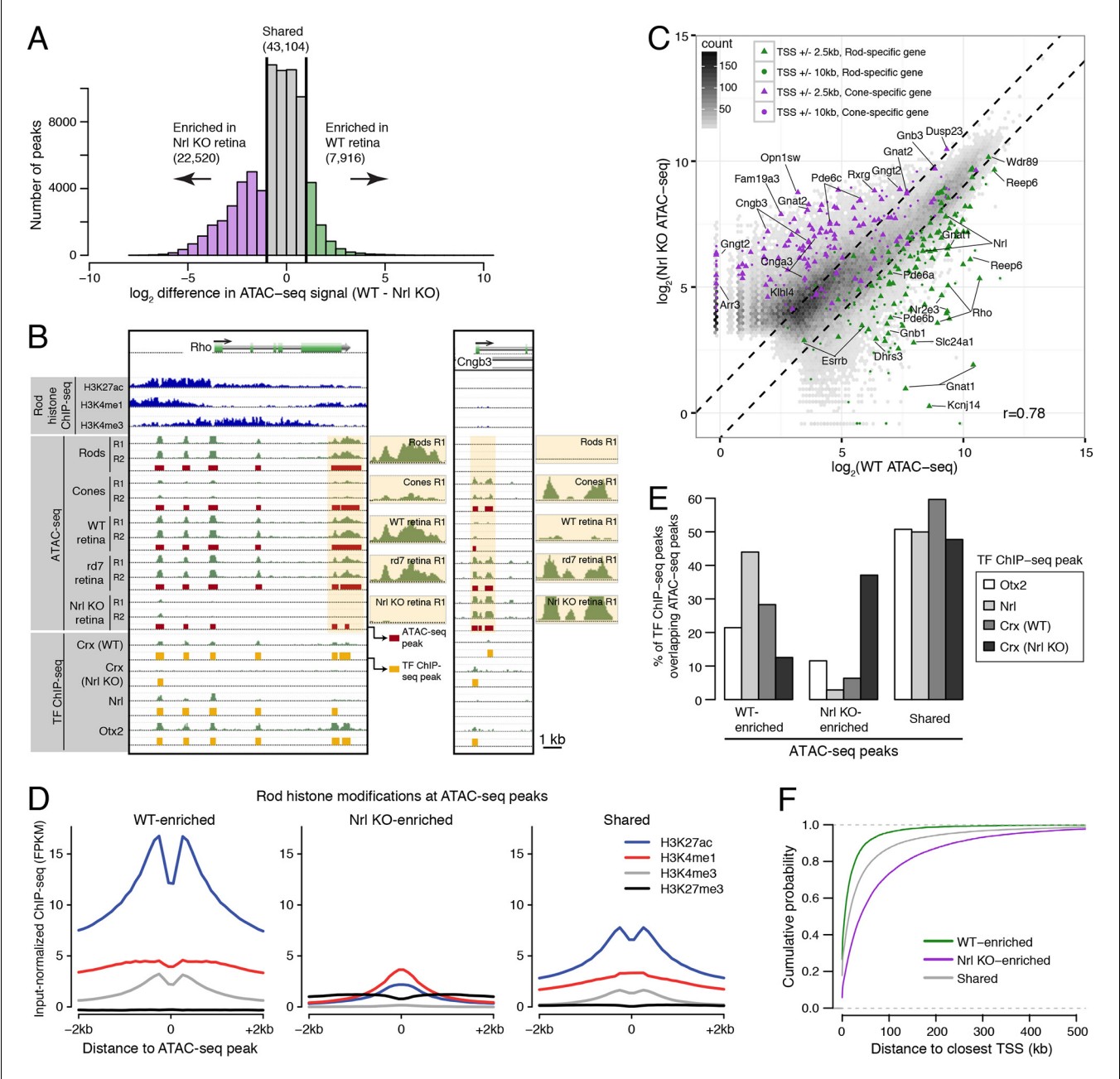

**Figure 3.** Distinctive features of WT-enriched versus *Nrl KO*-enriched accessible chromatin. (**A**) Histogram showing that the WT retina has nearly three-fold fewer number of enriched ATAC-seq peaks compared to the *Nrl KO* retina. (**B**) Browser images showing histone modification ChIP-seq signals (rods, top), ATAC-seq signals (middle), and TF ChIP-seq signals (bottom) near *Rho*, a rod-specific gene (left) and near *Cngb3*, a cone-specific gene (right). Enlarged images of the ATAC-seq signals in the highlighted area are shown for one replicate of each cell or tissue type. Bars below the raw data indicate locations identified as ATAC-seq peaks or TF ChIP-seq peaks. (**C**) Peaks near rod genes (green; e.g., *Nrl, Gnat1*) generally show higher ATAC-seq signals in WT than in *Nrl KO* retina. Peaks near cone genes (purple; e.g., *Pde6h, Pde6c*) generally show higher ATAC-seq signals in *Nrl KO* than in WT retina. Colored points show all ATAC-seq peaks that fall within 2.5 kb (triangle) or 10 kb (circle) of the TSS. Selected peaks are labeled by their associated gene. r, Pearson correlation. (**D**) Line plots showing that WT-enriched ATAC-seq peaks have higher mean levels of active rod histone modifications (H3K27ac, H3K4me1, and H3K4me3) compared to *Nrl KO*-enriched peaks. (**E**) Barplot showing the percentage of TF ChIP-seq peaks that overlap each category of ATAC-seq peak. (**F**) WT-enriched ATAC-seq peaks are distributed closer to the TSS than *Nrl KO*-enriched and shared ATAC-seq peaks.

The following figure supplements are available for figure 3:

**Figure supplement 1.** Comparisons of ATAC-seq signals between purified rod and cone nuclei.

*Figure 3 continued on next page*

*Figure 3 continued*

**Figure supplement 2.** ATAC-seq signals between biological replicates.

**Figure supplement 3.** Cell type-specific differences in ATAC-seq peak distribution are not reflected by gene expression.

ATAC-seq peaks. Importantly, low levels of ATAC-seq signals near rod-specific genes in the *Nrl KO* retina could not have originated from sample contamination by rods. These observations are in line with previous studies showing that rod-specific TFs NR2E3 and NRL bind to regulatory elements near both rod- and cone-specific genes (*Peng et al., 2005*; *Peng and Chen 2005*; *Oh et al., 2007*; *Onishi et al., 2009*), a phenomenon that presumably reflects a shared photoreceptor identity. Our data generalize previous results by showing shared chromatin accessibility, regardless of any particular TF, around photoreceptor genes. Our data further highlight large differences in the magnitudes of chromatin accessibility between WT and *Nrl KO* retinas at these regions.

Interestingly, ATAC-seq peaks in WT retina are distributed closer to promoters than peaks in *Nrl KO* retina (*Figure 3—figure supplement 3A*). WT retina-enriched peaks are also depleted in distal intergenic regions: only 4% (324) of WT-enriched peaks are >100 kb from a TSS, compared with 27% (5997) of *Nrl KO*-enriched peaks (*Figure 3F*; *Figure 3—figure supplement 3B*). These differences in chromatin accessibility do not appear to be associated with large differences in the overall distributions of gene expression levels (*Figure 3—figure supplement 3C*). Instead, these results raise the possibility that gene expression is regulated by more promoter-proximal sequences in rods compared with cones.

## DNA sequence determinants of accessible chromatin predict retinal enhancer activity

To test experimentally whether putative regulatory regions showed cell type-specific activity, we used in vivo retinal electroporation to ask whether discrete DNA segments that overlap ATAC-seq peaks could induce reporter activity in WT or *Nrl KO* retinas (*Figure 4*; *Figure 4—figure supplement 1*). Electroporation at postnatal day 0 (P0) into WT retina, where rod progenitors are actively proliferating, evaluates whether a DNA segment is active in rods; similarly, P0 electroporation into *Nrl KO* retina evaluates activity in S-cones (*Matsuda and Cepko, 2004*; *Swaroop et al., 2010*). We cloned DNA segments (mean length 552 bp; *Supplementary file 6*) located near rod genes (9 regions) or cone genes (16 regions) upstream of a minimal promoter and a GFP reporter (*Billings et al., 2010*), co-electroporated each construct together with a constitutively active tdTomato (TdT) control, and evaluated the native GFP fluorescence at TdT+ regions.

Six regions showed reporter activity whereas no signal was detected in the remaining regions. Lack of reporter signal may be due to the limited sensitivity of the native GFP fluorescence; furthermore, regulatory regions that function in a combinatorial manner may be missed (*Corbo et al., 2010*). Within the sensitivity limits of the assay, one previously identified enhancer near *Nr2e3* (*Hsiau et al., 2007*) induced reporter activity in WT but not in *Nrl KO* retinas (*Figure 4*, top), and four out of five regions near cone-specific genes induced reporter activity in *Nrl KO* but not in WT retinas (*Figure 4*, bottom; *Figure 4—figure supplement 1*). The fifth region (near *Thrb*) induced activity in both retina types. Although the electroporation experiments tested only a limited number of regions, the majority (5/6) of regions that had detectable reporter expression showed cell type-specificity.

To further explore how DNA sequences could reflect rod versus cone identity, we asked whether particular sequence features were preferentially found in regions with cell type-specific accessible chromatin. We first set stringent thresholds for both cell type-specific and shared accessible chromatin and defined 88 WT-specific, 1493 *Nrl KO*-specific, and 2463 shared 500 bp, non-promoter peaks. Consistent with our previous analysis (*Figure 3C*), WT-specific and *Nrl KO*-specific ATAC-seq peaks are preferentially located near rod-enriched and cone-enriched genes, respectively, and show the expected pattern of CRX, NRL, and OTX2 binding (*Figure 5—figure supplement 1A,B*).

We used MotifSpec (*Karnik and Beer, 2015*) in order to detect single *de novo* motifs that can discriminate between two sequence sets. Consistent with CRX binding data (*Figure 5—figure*

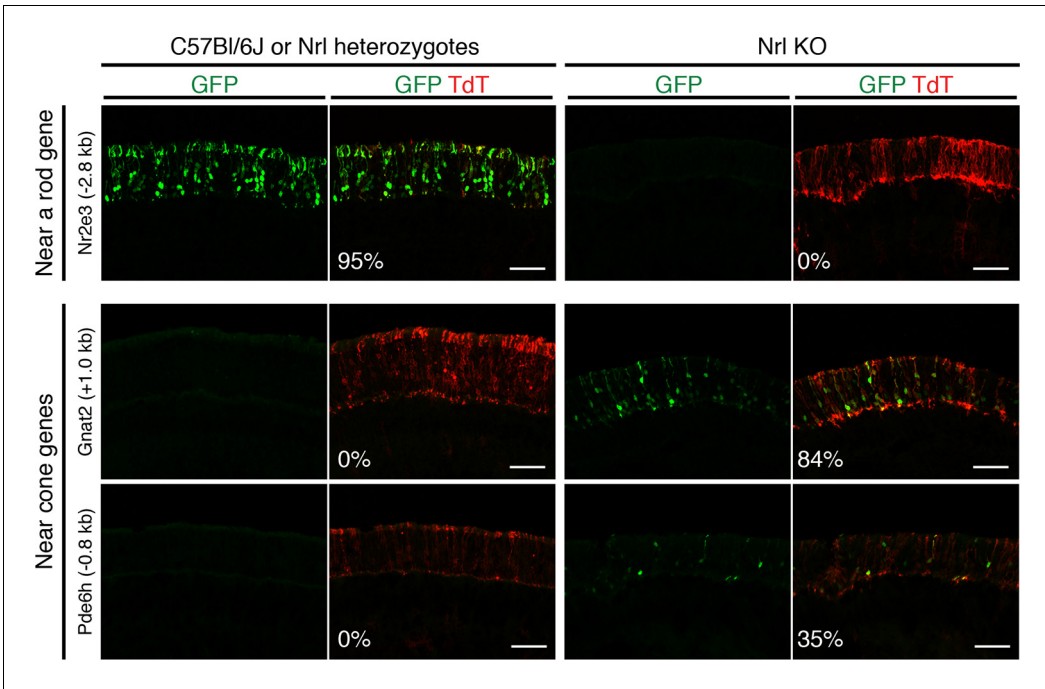

**Figure 4.** In vivo retinal electroporation of putative regulatory elements. Cryosections of C57Bl/6J or *Nrl* heterozygote retinas (left) and *Nrl KO* retinas (right) from 3–4 week old mice after in vivo retinal electroporation at P0 of a putative rod regulatory element near *Nr2e3* (top row) or putative cone regulatory elements near cone-specific genes (bottom rows). The element near *Nr2e3* induces GFP reporter expression only in WT retina but not in *Nrl KO* retina. Elements near *Gnat2* and *Pde6h* induce GFP reporter expression in *Nrl KO* retina but not in WT retina. The TdT signal is a control for electroporation efficiency. The average % of electroporated (TdT+) cells that are GFP+ is shown. Coordinates of electroporated elements are listed in *Supplementary file 6*.

The following figure supplement is available for figure 4:

**Figure supplement 1.** Additional examples of in vivo retinal electroporation of putative cone regulatory elements.

---

*supplement 1B*), a motif matching the canonical CRX motif ($p < 1 \times 10^{-10}$; *Gupta et al., 2007*) is enriched in all three peak sets (WT-specific, *Nrl KO*-specific, and shared) relative to random sequences (*Figure 5A*, left) and is the top motif for both *Nrl KO* retina and shared peaks. A motif matching CTCF ($p < 1 \times 10^{-5}$) shows the second strongest enrichment in shared peaks against random sequences (*Figure 5A*, middle). Shared peaks with the strongest CTCF binding sites appear to be located in accessible chromatin in a broad range of cell types: 87% of ATAC-seq peaks with the strongest (top 20%) CTCF binding sites show DNaseI hypersensitivity in >30% of mouse tissues surveyed by ENCODE (*Stamatoyannopoulos et al., 2012*), compared to only 35% of peaks in the remainder (bottom 80%) that lack strong CTCF binding sites. For WT-specific peaks, we report a novel motif (*Figure 5A*, right) that can distinguish this set from random sequences (area under receiver operating characteristic curve, auROC = 0.70). Although the biological significance of this motif remains unclear, we find that about half (33/72) of WT-specific peaks with strong CRX scores also contain high scores for this motif. Conversely, the majority of peaks (39 peaks; 75%) with strong scores for this novel motif also contain strong scores for CRX, but a sizeable minority (13 peaks; 25%) do not appear to be sites of CRX binding.

Because individual motifs are known to have relatively weak predictive power (*Ghandi et al., 2014*), state-of-the-art regulatory sequence prediction methods incorporate combinations of motifs for improved accuracy. Therefore, we next classified WT-specific, *Nrl KO*-specific, and shared peak sets against each other and against random sequences using a gapped *k*-mer support vector machine (gkm-SVM; *Ghandi et al., 2014*), which models TF binding specificity with a complete set of *k*-mer features (i.e., words of length *k*). Although the WT-specific set was too small for the gkm-

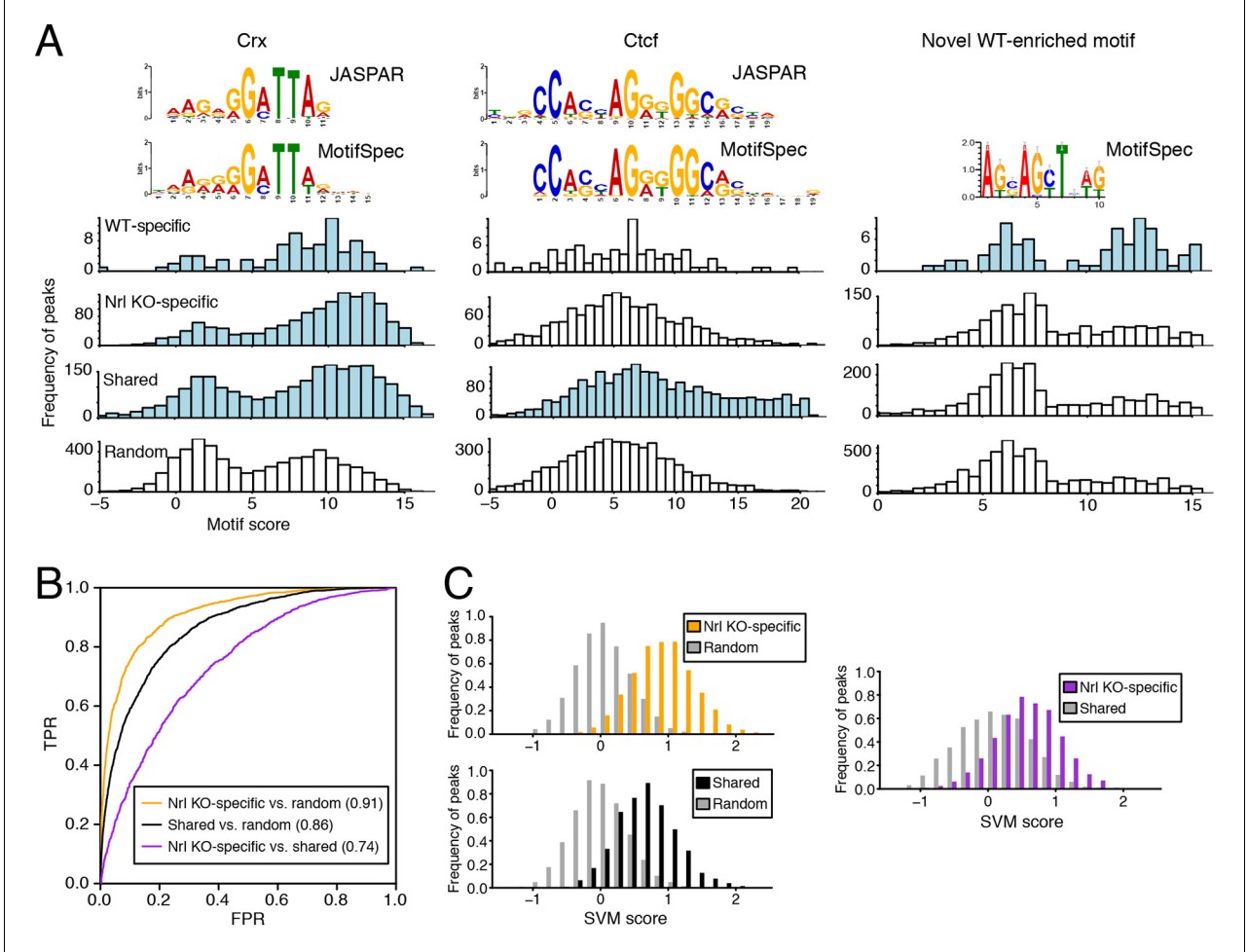

**Figure 5.** Machine learning identifies DNA sequence features of photoreceptor accessible chromatin. (**A**) Barplots showing score distributions of the strongest single motifs detected by a discriminative algorithm (MotifSpec) used to identify differentially enriched motifs. CRX binding sites are enriched in all sets of peaks relative to GC-matched random genomic sequences (left). CTCF is enriched at shared peaks (middle), and a novel motif is enriched at WT-specific peaks (right). In each peak set, the distribution of motif scores which are predictive above AUC = 0.6 versus random sequence is shown in blue. (**B–C**) ROC curves showing that gapped *k*-mer SVM can classify ATAC-seq peaks using regulatory sequence features (**B**). When trained versus GC-matched random genomic sequences, gkm-SVM auROC is high (in parentheses). Distinguishing *Nrl KO*-specific ATAC-seq peaks from shared peaks is more challenging (**C**, right) than distinguishing between *Nrl KO*-specific (**C**, top left) and shared (**C**, bottom left) peaks from random regions. Nevertheless, the sequence-based SVM score, a weighted sum of *k*-mer counts, is still able to distinguish *Nrl KO*-specific peaks from shared peaks based on sequence features.

The following figure supplements are available for figure 5:

**Figure supplement 1.** Retinal *k*-mers.

**Figure supplement 2.** DNA regulatory sequences inferred from retinal chromatin accessibility yield gkm-SVM scores which predict enhancer activity in a massively parallel reporter assay.

SVM analysis, the *Nrl KO*-specific (auROC = 0.91) and shared (auROC = 0.86) regions can be distinguished from random regions based on regulatory sequences contained within the peaks (*Figure 5B,C*). We find that both *Nrl KO*-specific and shared classes have large SVM weights for *k*-mers matching the CRX binding site (GATTA) (*Figure 5—figure supplement 1C*). Therefore, the relatively lower accuracy in classifying *Nrl KO*-specific peaks versus shared peaks (auROC = 0.74) may reflect the close developmental paths of rods and cones, which include their usage of common photoreceptor TFs. Even in this case, however, we find sequence features that allow the SVM score to

separate many of the *Nrl KO*-specific peaks (TTAA-enriched homeodomain binding sites) from shared peaks (CTCF binding sites).

Further supporting the ability of our gkm-SVM model to predict enhancer activity, we find that a gkm-SVM trained on WT retina ATAC-seq could predict retinal enhancer activity as assessed by a massively parallel reporter assay (*Shen et al., 2016*). In this assay, candidate *cis*-regulatory elements (CREs) from retina, brain, heart, and liver were joined to a minimal promoter and a barcoded transcription unit. Enhancer activity was tested in three independent experiments by introducing the CRE library (~45,000 barcodes covering >3000 CREs) into neonatal mouse retina, composed of primarily rods, followed by quantification of barcode abundances in the resulting RNA population. Using the number of replicates in which RNA was detected to discretize the expression level (*Figure 5—figure supplement 2A*), we find that gkm-SVM trained on WT retina ATAC-seq is a strong predictor of expression level. Among the 12858 constructs with RNA detected in at least one replicate, the Pearson correlation between the gkm-SVM WT retina ATAC-seq score and the average retinal CRE expression level is 0.427 (p<10$^{-320}$). Relative to all candidate CREs, those that score in the top 10% of gkm-SVM scores are 4-fold enriched for high-level expression (*Figure 5—figure supplement 2B*). Gkm-SVM trained on retina ATAC-seq performed slightly better than gkm-SVM trained on retina DNaseI hypersensitive sites (*Yue et al., 2014*), whereas training on chromatin features from unrelated cell types produced no enrichment. Conversely, candidate CREs that confer high-level expression have, on average, the highest gkm-SVM score when the model is trained on WT retina ATAC-seq regions (*Figure 5—figure supplement 2C*).

## Epigenomic patterns in *rd7* rods generally resemble WT rods, but also show features consistent with a partial rod-to-cone conversion

We next asked how perturbing rod development through loss of NR2E3 impacts the rod epigenome. Pearson correlations of ATAC-seq signal between *rd7* and WT retina (r = 0.91; *Figure 6A*) and between *rd7* and *Nrl KO* retina (r = 0.88; *Figure 6B*) are both higher than the correlation between WT and *Nrl KO* retina (r = 0.78; *Figure 3C*). These correlations indicate that the *rd7* rod chromatin shows a hybrid rod/cone phenotype, recapitulating previous observations from gene expression studies (*Chen et al., 2005*; *Corbo and Cepko, 2005*; *Peng et al., 2005*). Notably, 44 ATAC-seq peaks within 10 kb of rod-specific genes show greater than two-fold higher signal in WT retina compared to *rd7* retina, whereas only one peak displays the opposite pattern. Reciprocally, peaks near cone genes that are normally repressed in rods by NR2E3 (e.g., *Gnat2, Pde6c, Pde6h, Gnb3*) show higher chromatin accessibility in *rd7* retina compared to WT retina.

With respect to their DNA methylation, very few regions show differences between *rd7* rods and WT rods: 287 regions are hypo-DMRs in *rd7* rods and 1385 regions are hypo-DMRs in WT rods. A comparison of *rd7* rod versus cone methylomes reveals more differences: 1981 regions are hypo-DMRs in *rd7* rods and 4929 regions are hypo-DMRs in cones. Consistent with previous studies in the retina at targeted genes (*Merbs et al., 2012*) or using enrichment-based methylation assays (*Oliver et al., 2013*), intragenic and promoter levels of DNA methylation are lower in rods at rod-specific genes and in cones at cone-specific genes (*Figure 6C*). At rod-specific genes (e.g., *Nrl, Esrrb, Cnga1*), *rd7* rod methylation levels generally resemble those seen in WT rods or lie midway between WT rods and cones (*Figure 6C,D*; *Figure 6—figure supplement 1–2*). A rare exception to this pattern occurs at *Aqp1* (*Figure 6D*), a gene coding for an aquaporin water channel (*Papadopoulos and Verkman, 2013*). Both *rd7* rods and cones have low *Aqp1* RNA abundance and high DNA methylation at the *Aqp1* gene, whereas this gene is expressed and de-methylated in WT rods. In *rd7* rods, most cone-specific genes are methylated, except those that are de-repressed as a result of NR2E3 loss (*Figure 6C,E*; *Figure 6—figure supplement 1,3*).

The intermediate methylation levels at adult photoreceptor genes in *rd7* rods prompted us to ask whether these hybrid photoreceptors show evidence of a partial cell fate conversion at other types of genomic regions. In particular, we examined multi-kilobase, hypo-methylated domains termed DNA methylation valleys (DMVs) that are strongly enriched for TF genes (*Xie et al., 2013*) (*Figure 7A*). We categorized 782, 635, and 816 long (≥5 kb) UMRs as DMVs in WT rods, *rd7* rods, and cones, respectively. The large majority of these regions are located within 2.5 kb of the TSS (89–90%) and overlap at least one gene body (93–94%). DMV-associated genes are also, consistent with previous studies (*Xie et al., 2013*; *Mo et al., 2015*), highly enriched for DNA-binding factors (*Figure 7—figure supplement 1A*).

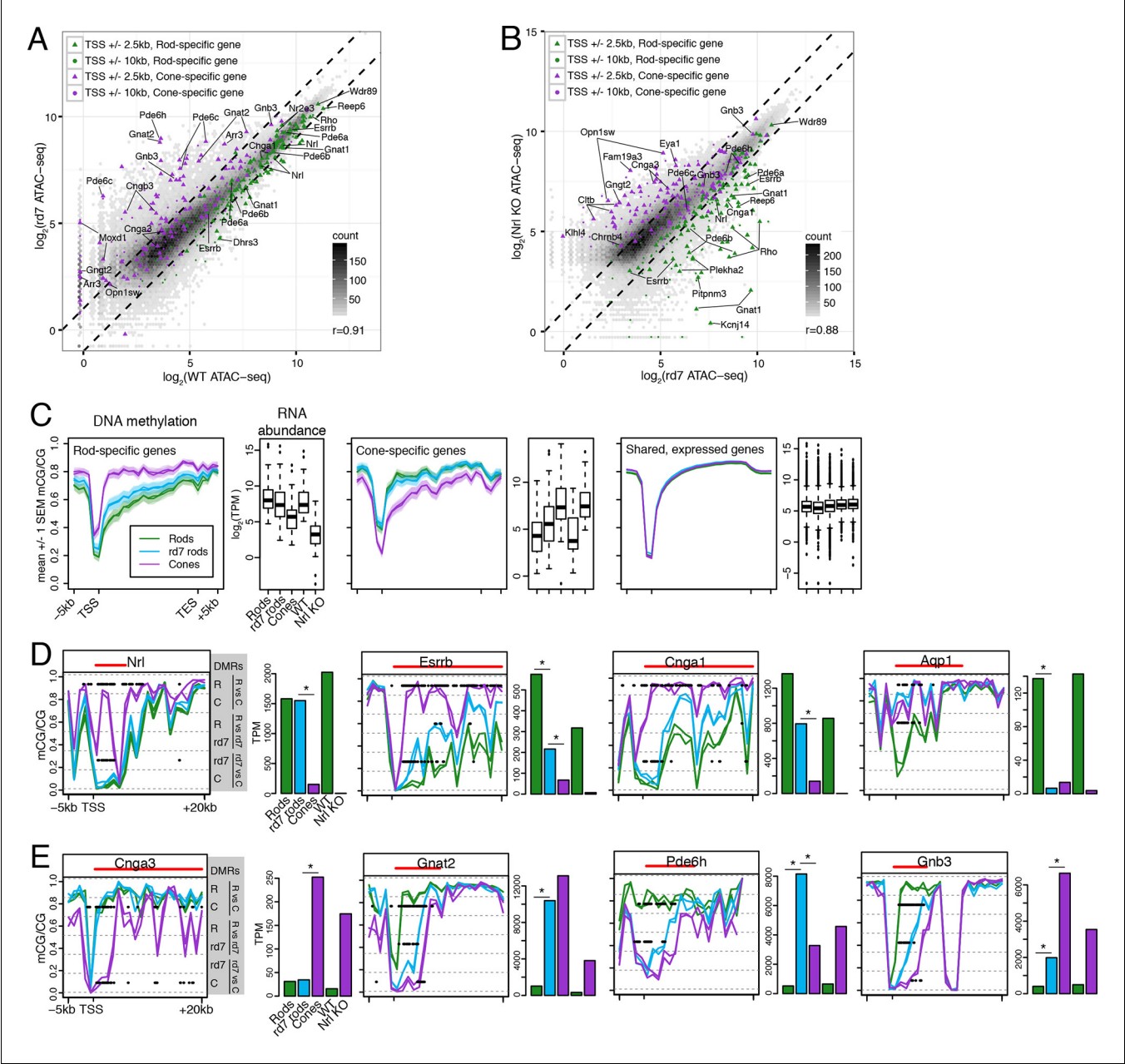

**Figure 6.** *rd7* rods show intermediate epigenomic profiles compared to WT rods and cones. (A–B) (A) Peaks near rod-specific genes (green) generally show equivalent ATAC-seq signals in WT and *rd7* retinas. A subset of peaks near cone-specific genes (purple) have higher signals in *rd7* retina than in WT retina. (B) Peaks near rod-specific genes (green) generally show higher ATAC-seq signal in *rd7* retinas than in *Nrl KO* retinas. Peaks near cone-specific genes (purple) show either similar ATAC-seq signals in *rd7* retina and *Nrl KO* retina or higher ATAC-seq signal in *Nrl KO* retina. For both (A) and (B), colored points show ATAC-seq peaks that fall within 2.5 kb (triangle) or 10 kb (circle) of a rod-specific gene (green) or a cone-specific gene (purple). Selected peaks are labeled by their associated gene. r, Pearson correlation. (C) Genes that are up-regulated in rods (left) and cones (middle) show lower levels of CG DNA methylation in rods and cones, respectively. SEM, standard error of the mean. (D–E) At individual rod-specific (D) and cone-specific (E) genes, line plots showing CG DNA methylation levels in a region between -5 kb and +20 kb around the TSS. Biological replicates are shown as separate lines (WT rods, green; *rd7* rods, blue; cones, purple). Pairwise DMRs are indicated with black lines. R, WT rods; C, cones; *rd7*, *rd7* rods. The gene body is indicated with a red line. Barplots showing RNA abundances. All genes are differentially expressed between WT rods and cones and between WT retina and *Nrl KO* retina. Asterisks indicate differentially expressed genes between *rd7* rods and WT rods or between *rd7* rods and cones.

The following figure supplements are available for figure 6:

**Figure supplement 1.** Epigenomic patterns of WT rods, *rd7* rods, and cones near photoreceptor genes.

*Figure 6 continued on next page*

*Figure 6 continued*
**Figure supplement 2.** CG DNA methylation around rod-specific genes.
**Figure supplement 3.** CG DNA methylation around cone-specific genes.

We further merged DMV coordinates to form a union of 996 regions (*Figure 7—figure supplement 1B*); of these regions, 425 overlap rod Polycomb-repressed (H3K27me3+; H3K4me3-) regions, and 483 overlap rod active (H3K27me3-; H3K4me3+) regions (*Supplementary files 7* and *8*). About half of the merged DMVs (508) show equal or higher methylation in cones (mean mCG = 12.0%) than in WT rods (6.7%), with *rd7* rods showing an intermediate methylation level (9.9%) (*Figure 7—figure supplement 1C*, left). This category includes H3K4me3+ DMVs overlapping *Nrl* and *Nr2e3*

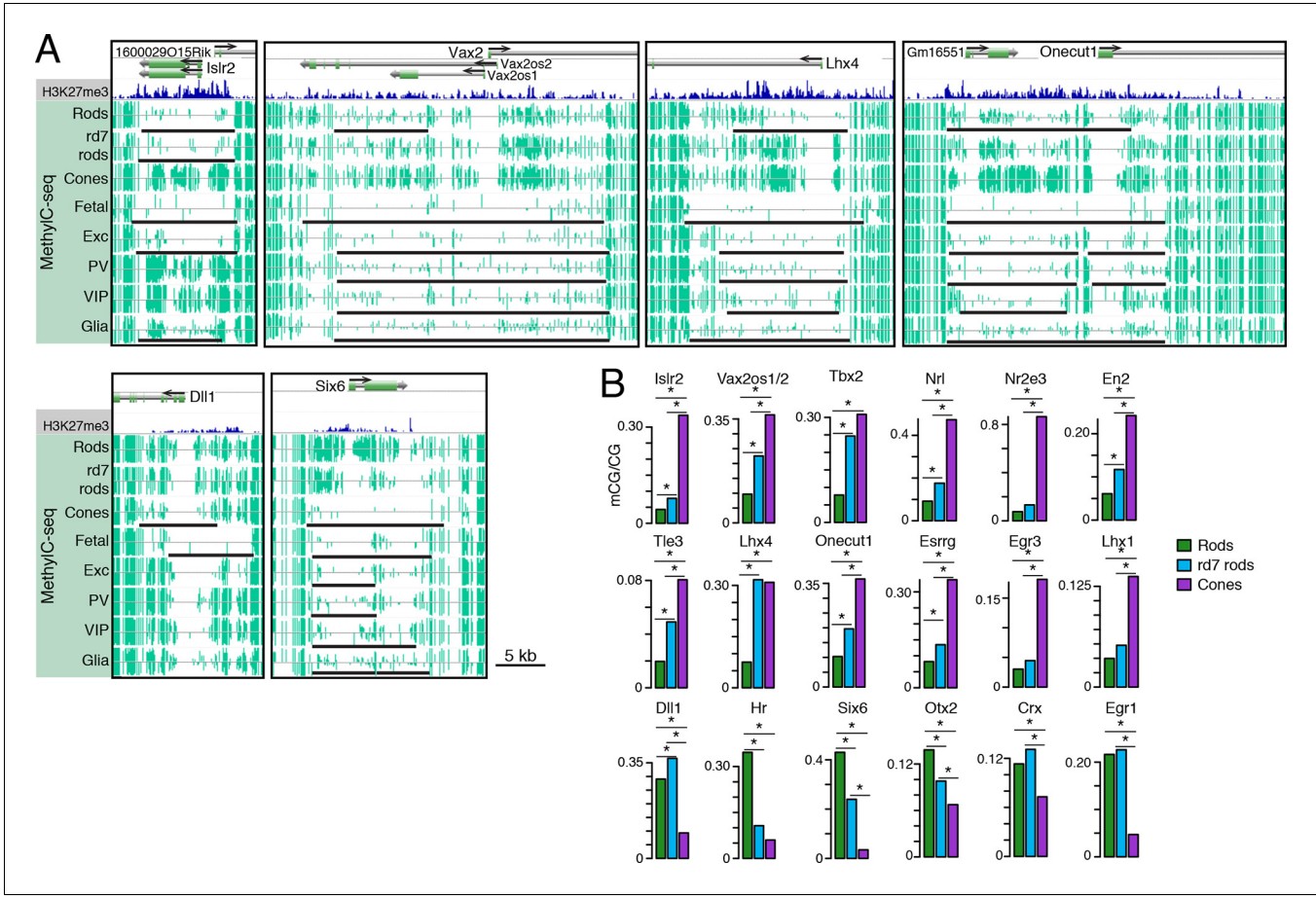

**Figure 7.** Retinal photoreceptors show distinct methylation patterns at DNA methylation valleys. (**A**) Browser images showing rod H3K27me3 (top track, blue) and CG methylation levels in retinal and cortical methylomes (bottom tracks, green). A variety of cell type-specific mCG patterns are shown at these regions, including hyper-methylation in all retinal samples compared to all cortical samples (e.g., *Vax2/Vax2os1/2*) and hyper-methylation in a subset of retinal and cortical samples (e.g., *Islr2*). Rod H3K27me3+ DMVs overlapping *Islr2, Vax2os1/2, Lhx4*, and *Onecut1* show higher levels of methylation in cones compared to rods. In contrast, rod H3K27me3+ DMVs overlapping *Dll1* and *Six6* show higher levels of methylation in rods compared to cones. Black lines indicate DNA methylation valleys identified in each cell type. (**B**) Barplots showing the levels of CG methylation in rods, *rd7* rods, and cones at DMVs overlapping individual TF genes. Asterisks indicate significance at FDR $< 1 \times 10^{-10}$ (Fisher's Exact Test).
The following figure supplement is available for figure 7:

**Figure supplement 1.** CG DNA methylation at DNA methylation valleys.

and H3K27me3+ DMVs overlapping *Islr2, Vax2os1/2, Lhx4*, and *Onecut1* (*Figure 7B*, top two rows). Interestingly, ONECUT1 is an early cone marker that drives cone genesis, acts upstream of NRL, and becomes silenced during cone maturation (*Emerson et al., 2013*; *Sapkota et al., 2014*). *Onecut1* overlaps a merged DMV that is 37% methylated in cones, 10% methylated in rods, and 20% methylated in *rd7* rods. This intermediate methylation level in *rd7* rods suggests that *Onecut1* may also have developmentally dynamic epigenomic or gene expression patterns in these hybrid photoreceptors.

The remaining 488 DMVs have higher mCG in WT rods than in cones; here, the mean methylation level is slightly higher in *rd7* rods (14.7%) than in WT rods (13.5%) (*Figure 7—figure supplement 1C*, right), which could potentially reflect a non-specific, genome-wide increase in DNA methylation in *rd7* rods (see *Figure 8A–C*). Certain individual DMVs, such as those overlapping *Hr, Six6*, and *Otx2*, show pronounced, stepwise decreases in DNA methylation across WT rods, *rd7* rods, and cones (*Figure 7B*, bottom row).

In summary, epigenomic patterns in *rd7* rods generally resemble those of WT rods, but may not attain the full magnitudes of rod chromatin accessibility or DNA hypo-methylation. At the same time, *rd7* rods also do not match the epigenomic patterns of cones. This hybrid epigenome highlights the importance of NR2E3 in activating and maintaining rod-specific photoreceptor identity as well as in repressing cone-specific attributes.

## A comparison of retinal photoreceptors to brain neurons reveals both local and global differences in DNA methylation patterns

Lastly, we compared the epigenomic landscapes of retinal photoreceptors with brain neurons. Non-CG methylation–referred to as mCH (where H=A, C, or T)–is a defining feature of brain neurons but is rare outside of neurons and pluripotent stem cells (*Xie et al., 2012*; *Lister et al., 2013*; *Schultz et al., 2015*). As retinal rods and cones are specialized sensory neurons, we asked whether they also have a high level of mCH. Compared to cerebral cortical neurons (*Mo et al., 2015*), rods and cones have similar levels of mCG but up to 29- and 7-fold lower levels of mCH, respectively (*Figure 8A–C*). However, rods and cones have higher mCH levels compared to most other non-brain tissue types (<0.05% mCH; *Schultz et al., 2015*). WT rods have lower mCH (0.11–0.13%) compared to cones (0.42–0.45%), whereas *rd7* rods show an intermediate mCH level (0.21–0.25%). Because NR2E3 is expressed after terminal mitoses in rods, our data also suggests that mCH accumulates post-mitotically in photoreceptors, as it does in the brain.

To further explore similarities and differences between retinal photoreceptors and cortical neurons, we quantified the epigenomic distance between samples by calculating the genome-wide Pearson correlation of DNA methylation at CG sites between all pairwise sample combinations (*Figure 8D*). Hierarchical clustering shows that retinal photoreceptors are tightly clustered, whereas cortical neurons cluster separately. In addition, cortical neurons show greater epigenomic distance between neuron subtypes compared to the rod-cone distance. A similar pattern is observed when epigenomic distance is calculated using chromatin accessibility (*Figure 8—figure supplement 1*).

Based on their DNA methylation patterns, the fetal cortex clusters more closely to mature retinal photoreceptors than to mature cortical neurons (*Figure 8D*). This clustering organization exists despite the anatomical difference between the cerebral cortex and retina and the development of many fetal cortical cells into mature cortical neurons. One interpretation of this clustering pattern is that cortical neurons may acquire more extensive cell type-specific modifications than photoreceptors during their developmental maturation. We therefore defined DMRs across all retinal and cortical samples and found twice as many regions that showed hypo-methylation only in retinal photoreceptors (63384) compared to those that showed hypo-methylation only in cortical neurons (31483; *Figure 8—figure supplement 2*, *Supplementary file 4*). Also consistent with our interpretation, a larger proportion of retinal hypo-DMRs, compared to cortical hypo-DMRs, display low-to-moderate levels of DNA methylation in fetal cortex (*Figure 8E*).

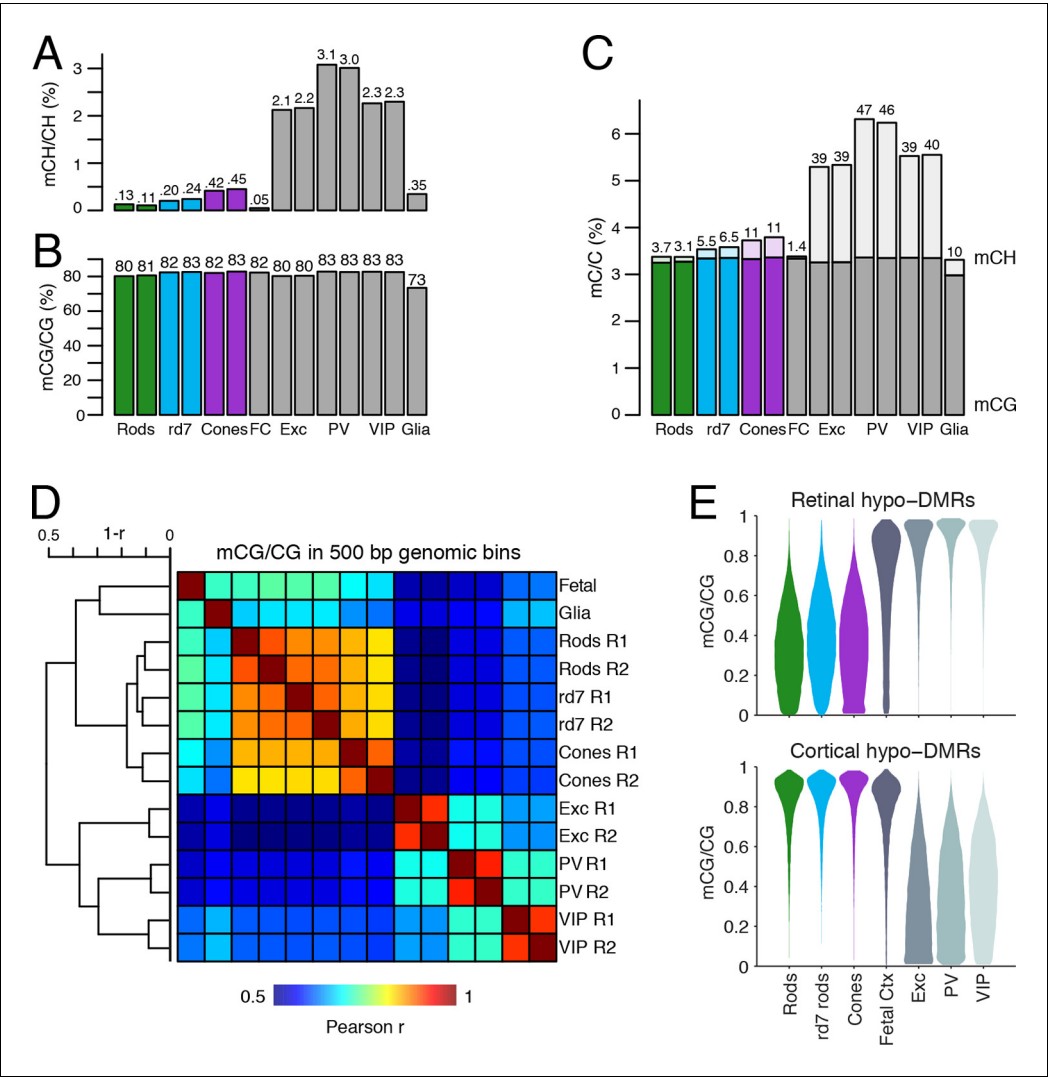

**Figure 8.** DNA methylation at retinal photoreceptors versus cortical neurons. (**A–B**) The levels of CH (**A**) and CG (**B**) DNA methylation for retinal and cortical cell types. FC, E13 fetal cerebral cortex. (**C**) The total level (CG and CH) of DNA methylation. The percentage of all methylcytosines that are in the CH context (top) is shown. (**D**) Heatmap showing the pairwise Pearson correlation (r) of CG methylation levels in 500 bp genomic bins among retinal and cortical samples. The dendrogram shows hierarchical clustering using 1-r as the distance measure. Biological replicates (R1, R2). (**E**) The fetal cortex shows a lower distribution of mCG/CG at pan-retinal hypo-DMRs (top) compared to pan-cortical hypo-DMRs (bottom).

The following figure supplements are available for figure 8:

**Figure supplement 1.** Correlations of accessible chromatin among retinal and cortical samples.

**Figure supplement 2.** Retinal versus cortical hypo-DMRs.

## Discussion

### A lower concordance between accessible chromatin and DNA methylation is correlated with increased chromatin condensation

In comparing rod and cone landscapes of DNA methylation and chromatin accessibility, we have uncovered several unusual features of rod photoreceptors. First, rods have relatively fewer regions of high chromatin accessibility, and on average, rod-enriched accessible chromatin sites are located

closer to promoters. Second, rods have more un-methylated and low-methylated regions compared to cones. Third, compared with cone hypo-DMRs, rod hypo-DMRs are located at greater distances from the TSS and show seven-fold lower overlap with ATAC-seq peaks. Furthermore, rod hypo-DMRs are enriched for regions that are both hypo-methylated and marked by active histone modifications in fetal neural tissue. These findings suggest that many hypo-methylated regions in rods may mark previously active fetal enhancers that have retained their hypo-methylation despite loss of enhancer activity (*Figure 9*). Such regions have previously been described as vestigial enhancers

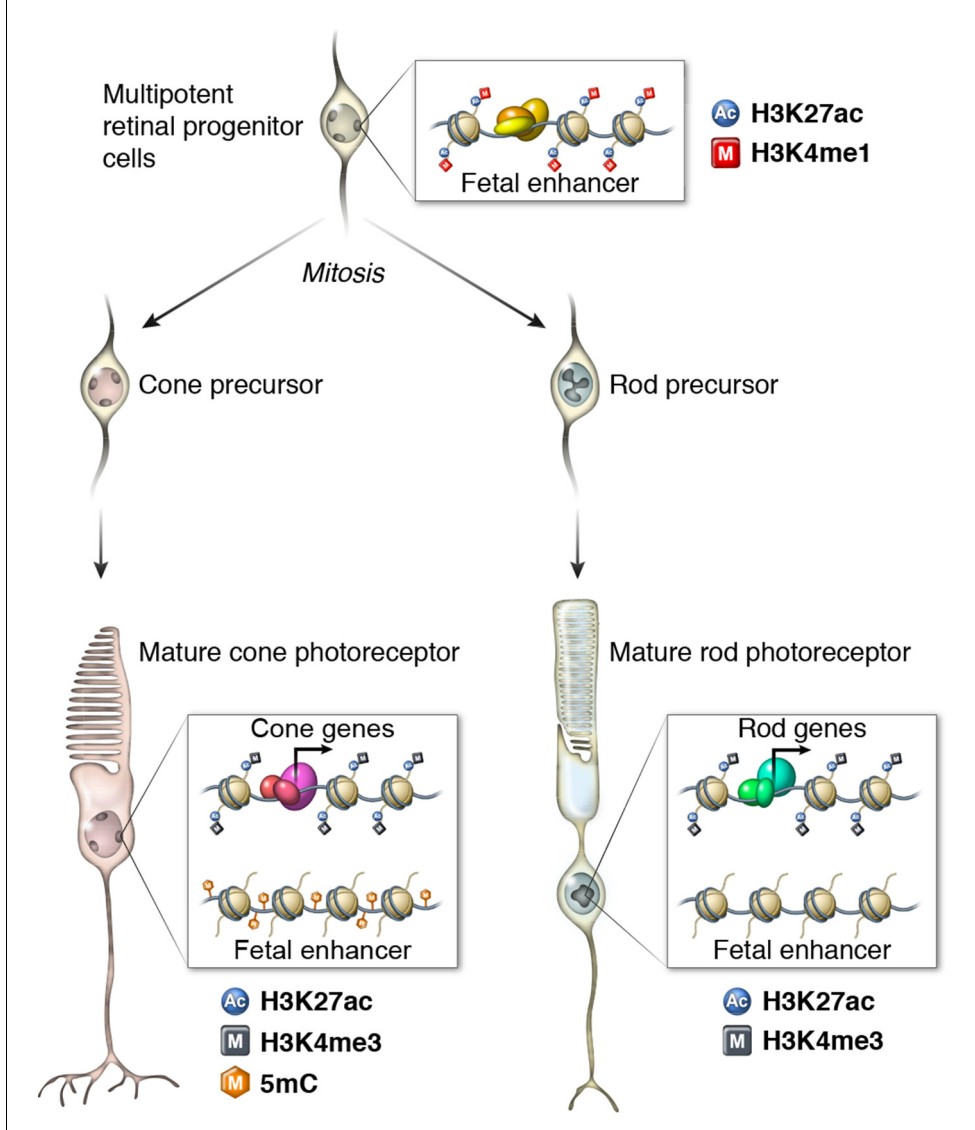

**Figure 9.** Epigenomic model of rod and cone photoreceptor development. Enhancers that are active only in progenitor cells (termed 'fetal-only', as the fetal brain was used as a rich source of generic neural progenitors) have low levels of DNA methylation and are enriched for H3K27ac and H3K4me1 histone modifications. In mature cones, histones near fetal-only enhancers lose H3K27ac and H3K4me1 and there is a gain of DNA methylcytosines. In contrast, in mature rods, fetal-only enhancers lose H3K27ac and H3K4me1 but the DNA remains unmethylated, potentially due to the barrier to cytosine methyltransferases posed by their high level of chromatin condensation. In both rods and cones, expressed genes, including rod- and cone-specific photoreceptor genes, have promoters marked by low DNA methylation, high chromatin accessibility, and enrichment for H3K27ac and H3K4me3. Active enhancers are marked by low DNA methylation, high chromatin accessibility, and enrichment for H3K27ac and H3K4me1 (not shown).

(*Hon et al., 2013*), but the factors that contribute to continuing hypo-methylation and to variation in the number of vestigial enhancers across different cell types remained unclear.

Mammalian cells show a spectrum of nuclear size, and rod nuclei fall near one extreme of this spectrum (*Solovei et al., 2009*). Mouse rod nuclei are exceptionally small, and increased chromatin condensation in rods could potentially pose a barrier to DNA methylation by limiting the accessibility of DNA methyltransferases. In support of this hypothesis, we find that *rd7* rods, a hybrid rod/cone cell with slightly increased nuclear size and lower levels of chromatin condensation compared to rods (*Corbo and Cepko, 2005*), show higher levels of DNA methylation at putative rod vestigial enhancers. *rd7* rods also show intermediate levels of mCH relative to normal rods and cones. A previous study showed that dense heterochromatin regions surrounding immunoglobulins or olfactory receptor clusters in brain neurons coincide with regions of low mCH (*Lister et al., 2013*), consistent with the idea that the highly condensed chromatin in rods could be a global limiting factor to mCH accumulation. Taken together, our results suggest that high chromatin compaction could provide a unifying explanation of some of the unique epigenomic features of rod photoreceptors. These observations may be relevant to other cell types with different nuclear sizes and extents of chromatin condensation. For example, cells in the oligodendrocyte lineage undergo chromatin compaction as they progress from progenitors to mature myelinating oligodendrocytes (*Shen et al., 2005*).

## The epigenome as a reflection of cellular development

A progressive restriction of the accessible chromatin landscape is a hallmark of differentiating cells and can be conceptualized with a genome-centric version of the Waddington landscape of cell differentiation (*Waddington, 1940*; *Stergachis et al., 2013*). In this landscape, a ball rolls down a succession of narrowing valleys. The ball represents a differentiating cell traveling along a trajectory that leads to a progressively restricted fate. Our finding that rods show fewer accessible chromatin regions than cones suggests that, at the epigenomic level, a rod could represent a more developmentally restricted cell type than a cone. This idea is consistent with current models of photoreceptor genesis, which propose that NRL, perhaps in combination with NR2E3, is necessary to direct photoreceptor precursors to turn off the default S-cone fate and differentiate along the rod lineage (*Swaroop et al., 2010*).

We find that perturbing normal rod development by inactivating NR2E3 converts the rod epigenome to a state that is intermediate between that of rods and cones. *rd7* rods fail to attain WT rod levels of chromatin accessibility and DNA hypo-methylation at rod-specific regions, and they exhibit increased accessibility and hypo-methylation at multiple cone-specific regions. Furthermore, *rd7* rods show intermediate patterns of DNA methylation at large hypo-methylated domains that overlap key developmental TF genes. Future studies at multiple developmental timepoints will be necessary to explore how chromatin accessibility and DNA methylation evolve differentially in WT rods, *rd7* rods, and cones, particularly at the point of photoreceptor fate commitment and early differentiation.

## Clinical relevance of rod and cone epigenomes

In addition to enriching our understanding of photoreceptor gene regulation, mapping the epigenomic landscape of rods and cones could be informative for understanding the clinical variability of human retinal diseases. In inherited retinal diseases such as retinitis pigmentosa and Stargardt macular dystrophy, patients with the same coding region mutation can differ markedly in the age of onset, rate, and pattern of photoreceptor degeneration (*Shastry, 1994*; *Fishman et al., 1999*; *Oh et al., 2004*). Although part of this phenotypic heterogeneity may be a result of environmental factors such as light exposure (*Wright et al., 2010*), another contributing factor could be differences in regulatory DNA sequences that would affect the binding of TFs. Therefore, characterizing genome-wide accessible chromatin in rods and cones could identify regions that influence the clinical course of retinal disease.

The identification of rod and cone regulatory regions could also be useful in the quest to develop cell-based therapies to treat retinal disease. Recent advances in cellular reprogramming have led to the production of stem cell-derived retinal photoreceptors that might be transplanted into the diseased retina to restore vision (*Lamba et al., 2009*; *Gonzalez-Cordero et al., 2013*). Our datasets and integrative analyses could help in refining the reprogramming approaches in current use by

identifying key regulatory regions that drive photoreceptor identity, as well as by providing benchmarks for assessing the extent to which the reprogramming process has faithfully recapitulated a normal rod or cone identity.

## Materials and methods

### Mice

We crossed *Lmopc1-Cre* (*Le et al., 2006*) and *HRGP-Cre* mice (*Le et al., 2004*) with INTACT mice (*R26-CAG-LSL-Sun1-sfGFP-myc*; *Mo et al., 2015*) to generate progeny with tagged rod and cone nuclei, respectively. *Lmopc1-Cre; R26-CAG-LSL-Sun1-sfGFP-myc* and *HRGP-Cre; R26-CAG-LSL-Sun1-sfGFP-myc* retinas appeared morphologically normal with no retinal degeneration. For purified rods and cones, we used mice between 8–11 weeks of age.

We also used WT C57BL/6J mice (JAX 000664), *rd7* mice (JAX 004643), and *Nrl KO* mice (*Mears et al., 2001*). Both *rd7* and *Nrl KO* mice were congenic on a C57BL/6J background. For ATAC-seq, we used WT, *rd7*, and *Nrl KO* mice aged between P21 and P25. Relative to the mice used for INTACT, a younger set of mice were used in the whole retina ATAC-seq experiments in order to minimize effects due to low-level retinal degeneration in the *Nrl KO* retina (*Mears et al., 2001*) and *rd7* retina (*Haider et al., 2001*). For all sequencing experiments, only male mice were used.

### Immunohistochemistry

Standard procedures for immunohistochemistry were used. Retinas were sectioned at 10 μm thickness using a cryostat. Whole-mount retinas were prepared by fixing eyes with 1.5% PFA for 1 hr at room temperature before dissection and immunohistochemistry. The following reagents were used: DAPI, rabbit anti-GFP (1:400, A11122, Life Technologies, Carlsbad, CA); rhodamine labeled Peanut Agglutinin (1:1000, RL-1072, Vector Laboratories, Burlingame, CA); rabbit anti-GFAP (1:200, RB-087-A, NeoMarkers, Fremont, CA); and Alexa Fluor-conjugated IgG secondary antibodies (1:400, Life Technologies).

### Microscopy

Confocal images were taken with an LSM700 (Zeiss, Jena, Germany) microscope. Image processing was performed using Adobe Photoshop (Adobe Systems Inc., San Jose, CA), and included adjustments of brightness, contrast, and levels in individual channels for merged color images. For electroporation images, identical settings were maintained between WT and *Nrl KO* retinas.

### Flow cytometry

For each experiment, retinas from two to four *HRGP-Cre; R26-CAG-LSL-Sun1-sfGFP-myc* mice were homogenized using a loose pestle in a Dounce homogenizer in ice-cold homogenization buffer (0.25 M sucrose, 25 mM KCl, 5 mM MgCl$_2$, 20 mM Tricine-KOH, 1 mM DTT, 0.15 mM spermine, 0.5 mM spermidine) with EDTA-free protease inhibitor (11 836 170 001, Roche, Basel, Switzerland). After addition of 5% IGEPAL-630 to bring the sample to 0.3% IGEPAL-630, the sample was further homogenized using a tight pestle. The sample was filtered using a 40 μm strainer (08-771-1, Fisher Scientific, Waltham, MA), mixed with 1.5 ml of 50% iodixanol density medium (D1556, Sigma, St. Louis, MO), and pelleted by centrifugation at 10,000g for 18 min in a swinging bucket centrifuge at 4°C. Nuclei were sorted using a MoFlo MLS high-speed cell sorter (Beckman Coulter, Brea, CA). Nuclei were sorted into either Buffer RLT for RNA preparation or PBS for DNA preparation (see below). An aliquot of nuclei sorted using the same parameters was placed into an additional tube. After sorting, this aliquot was inspected using a Zeiss Imager Z1 and Apotome system in order to verify the purity of the sorted sample.

### INTACT

INTACT purification of rod and cone nuclei was performed as previously described (*Mo et al., 2015*) with the modification that retinas were homogenized in 1.5 ml of homogenization buffer, and 1.5 ml (instead of 5 ml) of 50% iodixanol density gradient was added to the sample. In contrast to brain homogenates, which consisted predominantly of singlet nuclei, retinal homogenates showed a

mixture of singlet, doublet, and higher order nuclear aggregates. The inability of Dounce homogenization to fully dissociate retinal nuclei into a suspension of single nuclei was presumably due to the tight packing of retinal photoreceptors and the small size of photoreceptor cell bodies. Based on fluorescence microscopy of purified nuclei (>400 nuclei/experiment), INTACT purification of rod photoreceptors was 97.5% specific (96.7–98.2%; n = 5), with non-rod nuclei nearly exclusively arising from non-singlet aggregates.

## In vivo electroporation of reporters

Candidate rod- and cone-specific regulatory elements and controls (*Supplementary file 6*) were cloned into the Stagia3 plasmid (*Billings et al., 2010*), which consists of a minimal promoter upstream of eGFP-IRES-PLAP. All DNA segments overlapped ATAC-seq peaks. DNA segments were selected based solely on their location relative to known photoreceptor genes and not by measures of mammalian sequence conservation, ATAC-seq peak intensity, or differential ATAC-seq signal. Stagia3 plasmids (5 µg/µl) were co-electroporated together with a CMV-driven MARCKS-TdTomato plasmid (1 µg/µl) into C57Bl/6J, *Nrl* heterozygous, and *Nrl KO* retinas. In vivo electroporation into P0 mouse retina was performed as previously described (*Matsuda and Cepko, 2004*). Eyes were harvested at 3–4 weeks and immersion-fixed for 1 hr in 1.5% PFA at room temperature. Dissected retinas were equilibrated in 30% sucrose, embedded in OCT, and sectioned at 10 µm thickness using a cryostat. Slides were stained with DAPI and mounted with Fluoromount-G (0100–01, SouthernBiotech, Birmingham, AL). The native GFP and TdT fluorescence was viewed with a Zeiss LSM700 confocal microscope. Native fluorescence was chosen because of the relative linearity of the signal compared to alkaline phosphatase staining or immunostaining; however, due to its lower sensitivity, weak reporter activity was not detected. If a construct showed enhancer activity in either WT retina or *Nrl KO* retina, enhancer strength was quantified by counting the number of TdT+ nuclei that were also positive for GFP signal. For each construct and genotype (i.e., WT or *Nrl KO*), over 100 nuclei were counted across three experimental replicates, except for the *Opn1sw* -1 bp construct in WT retina, where only two replicates were used.

## Sample preparation

RNA, DNA, and nucleosomes from INTACT-purified nuclei were prepared as described in *Mo et al., 2015*. Briefly, whole RNA was prepared using the RNeasy Micro kit (74004, Qiagen, Venlo, Netherlands) with on-column DNase digestion. For RNA purification from FACS-sorted cone nuclei, nuclei were directly sorted into a tube containing Buffer RLT (Qiagen 74004). DNA was prepared using the DNeasy Blood and Tissue kit (Qiagen 69504). For DNA purification from FACS-sorted cone nuclei, nuclei were sorted into a tube containing PBS. Nucleosomes for native ChIP-seq were prepared by digesting 1–2 million nuclei with 0.025 units/µl micrococcal nuclease (LS004798, Worthington, Lakewood, NJ) at 37°C for 15 min.

## Library preparation and sequencing

Libraries for RNA-seq, MethylC-seq, ChIP-seq, and ATAC-seq were prepared as previously described, with slight modifications (*Garber et al., 2012*; *Buenrostro et al., 2013*; *Lister et al., 2013*; *Mo et al., 2015*). Briefly, total RNA was converted to cDNA and amplified (Ovation RNA-seq System V2, #7102, NuGEN Technologies Inc., San Carlos, CA). After adding a spike-in of ERCC RNA (4456740, Life Technologies), amplified cDNA was fragmented, end-repaired, linker-adapted, and sequenced for 50 cycles on a HiSeq 2500 (Illumina Inc., San Diego, CA). MethylC-seq libraries were PCR amplified with KAPA HiFi HotStart Uracil+ ReadyMix (KK2802, Kapa Biosystems, Wilmington, MA) and sequenced on an Illumina HiSeq 2000 up to 101 cycles. Histone ChIP-seq was performed by scaling down the HT ChIP-seq protocol (*Garber et al., 2012*). Each ChIP-seq reaction used chromatin prepared from 0.5–1 million nuclei, 25 µl Protein G Dynabeads (10004D, Life Technologies), and 1 µg of the following antibodies: rabbit anti-H3K27me3 (07–449, Millipore, Billerica, MA), rabbit anti-H3K27ac (ab4729, Abcam, Cambridge, UK), rabbit anti-H3K4me3 (ab8580, Abcam), and rabbit anti-H3K4me1 (ab8895, Abcam). Input and ChIP-enriched DNA was end-repaired, linker-adapted, amplified, and sequenced on an Illumina HiSeq 2500 for 50 cycles. ATAC-seq on 50,000 INTACT-purified nuclei was performed as in *Buenrostro et al. (2013)* with modifications as in *Mo et al. (2015)*. For each ATAC-seq sample using whole WT, *rd7*, and *Nrl KO* retinas, both retinas from one

mouse were homogenized in 1.5 ml of homogenization buffer, as described above. The homogenate was filtered through a 10 µm filter (04-0042-2314, Sysmex Partec, Kobe, Japan) into a 15 ml glass tube (Corning, Corning, NY). The homogenate was brought up to 5–6 ml with homogenization buffer and pelleted at 400 g for 10 min at 4°C. After rinsing the pellet once with 1 ml of homogenization buffer, the pellet was incubated with 50 µl of homogenization buffer on ice for 10 min with gentle pipetting to resuspend the nuclei. After quantifying the nuclei concentration using a hemocytometer, approximately 50,000 nuclei (up to 2.5 µl of the concentrated suspension) were used in a 50 µl Tn5 transposition reaction.

## Data analysis

### General

Basic data processing used BEDTools (*Quinlan and Hall, 2010*) and custom scripts. Reads were aligned to the mm10 genome using Bowtie (*Langmead et al., 2009*; *Langmead and Salzberg, 2012*) or Tophat (*Trapnell et al., 2009*). AnnoJ (*Lister et al., 2009*) was used to create browser representations.

For many parts of the analysis, unions of features (i.e., ATAC-seq peaks, TF ChIP-seq peaks, histone ChIP-seq peaks) across the two biological replicates were used. For example, we used the union of ATAC-seq peaks across replicates in order to examine its relationship with DNA methylation (*Figure 1B,D*; *Figure 1—figure supplement 2*; *Figure 2B*).

Non-overlapping random sampling of mappable genomic regions matching the sizes of ATAC-seq peaks (*Figure 1B*) or <5 kb UMRs and LMRs (*Figure 1C,E*) was performed using bedtools *shuffle*. Random sampling was performed ten times. Annotations of genome gaps and CG islands were obtained from the UCSC table browser. Bootstrap Kolmogorov-Smirnov test was performed in R (*ks.boot*, package 'Matching') using nboots = 1000.

### MethylC-seq

MethylC-seq data was first processed as described in *Schultz et al., 2015* and *Mo et al., 2015*. Unmethylated regions (UMRs) and low-methylated regions (LMRs) were identified using Methyl-SeekR (*Burger et al., 2013*) with m = 0.5 and 5% FDR and are listed in *Supplementary file 2*. For the analysis shown in *Figure 1* and *Figure 1—figure supplement 2*, only UMRs and LMRs <5 kb in length were used.

We identified differentially methylated regions (DMRs) using two approaches. First, we identified differentially methylated sites (DMS) across pairs of retina samples (namely, rods versus cones; rods versus *rd7* rods; and *rd7* rods versus cones). This approach used DSS (*Feng et al., 2014*), a beta-binomial distribution approach that incorporates replicate information in the modeling. DMSs were filtered with false discovery rate (FDR) <0.01. DMSs were combined into DMRs by joining those with FDR <0.01 into blocks if they had methylation differences in the same pairwise direction (e.g., rods < cones) and if they were within 250 bp of each other. Blocks overlapping DMSs in the opposite direction were removed. Then, DSS DMRs were defined as those blocks containing ≥2 significant differential CG sites.

For *Figure 2G*, each row represents a DMR between rods and cones, and each column represents a 1 kb bin in a 100 kb window around the closest TSS for the DMR. The column corresponding to the position of the DMR relative to the closest TSS was set to 1, with the rest of the elements set to 0. For plotting, the matrix was vertically smoothened with a sliding window size of 50 to allow the representation of global DMR spatial distributions.

Second, in order to identify DMRs between retina and cortex, we used Methylpy (*Schultz et al., 2015*). As we observed high consistency between biological replicates (*Figure 8D*), we pooled reads from pairs of replicates to increase the statistical power to detect DMRs. Six samples were included in the Methylpy algorithm: WT rod, *rd7* rod, and cone methylomes from this study, and excitatory, PV, and VIP cerebral cortical methylomes from *Mo et al. (2015)*. Then, retinal hypo-DMRs were defined by selecting those regions with hypo-methylation in both rods and cones but not in excitatory, PV, and VIP samples. Cortical hypo-DMRs were defined by selecting those regions with hypo-methylation in excitatory, PV, and VIP samples, but not in retinal rods and cones. Both DSS and Methylpy DMRs are shown in *Supplementary file 4*.

DNA methylation valleys (DMVs) were classified as those UMRs ≥5 kb with value 'mean.meth' ≤15. We then defined a new set of DMV regions by merging DMV coordinates across WT rods, *rd7* rods, and cones (*Supplementary file 7*). Fisher's exact test, followed by FDR correction, was used to determine statistical significance of differences in mCG/CG across WT rods, *rd7* rods, and cones at merged DMVs. DMVs were also categorized into those that overlapped H3K4me3 ChIP-seq peaks or H3K27me3 ChIP-seq peaks by ≥1 bp. GREAT analysis (*McLean et al., 2010*) to determine GO enrichment was performed using default settings. For the background, DMVs were combined with UMRs between 1–3.5 kb with value 'mean.meth' ≤15.

For *Figure 8A–C*, methylation levels were calculated for autosomes using the same procedure as in *Lister et al. (2013)* and *Mo et al. (2015)*, including adjustment for the rate of bisulfite non-conversion (listed in *Supplementary file 1*).

## ATAC-seq

ATAC-seq data was processed by trimming adapter sequences (*cutadapt* v1.3 -a CTGTCTCTTA TACACATCT -q 30 –minimum-length 36 –paired-output), aligning (BOWTIE2 v2.1.0 -t -X2000 –no-mixed –no-discordant), and then removing duplicate reads (*picard MarkDuplicates*). ATAC-seq peaks were called using paired-end reads that were less than 100 bp in length. Peaks were called using MACS2 (macs2 2.1.0.20140616 *callpeak* -p 0.00001 –call-summits –nomodel –shift -50 –extsize 100) (*Zhang et al., 2008*). Peaks for each replicate are shown in *Supplementary file 3* in MACS2 narrow-Peak format.

In order to categorize ATAC-seq peaks into those with similar or differing levels of accessibility in WT retina versus *Nrl KO* retina (*Figure 3*), in INTACT-purified rods versus cones (*Figure 3—figure supplement 1*), in WT retina versus *rd7* retina (*Figure 6A*), or in *rd7* retina versus *Nrl KO* retina (*Figure 6B*), we used DiffBind (*Ross-Innes et al., 2012*; *Stark and Brown, 2011*) with the DESEQ method (*Anders and Huber, 2010*). For each comparison, DiffBind first generates a set of consensus peaks using the requirement that peaks must be in at least two of the samples (minOverlap = 2). Log$_2$-normalized read counts in each sample are then tabulated over each peak region and plotted. To retrieve a set of high-confidence cell type-enriched peaks, we used consensus peaks with absolute fold difference >2 and FDR <0.01 as differentially enriched peaks. Peaks with absolute fold difference ≤2 were classified as shared peaks.

## ChIP-seq

For processing of retina TF and histone modification ChIP-seq, we aligned reads to the genome (BOWTIE v0.12.7 -m 1) and removed redundant reads. To examine rod H3K4me1, H3K4me3, H3K27ac, and H3K27me3 ChIP-seq signal over WT-enriched, *Nrl KO*-enriched, and shared ATAC-seq peaks (*Figure 3D*), replicates for each modification were combined. Reads were extended up to 150 bp in the 3' direction before calculating coverage in 100 bp windows (bedtools *slopBed* and *coverageBed*) and normalizing for library size. Input tracks were processed in the same way and subtracted from the ChIP-seq signals.

To identify TF ChIP-seq peaks, we obtained previously published ChIP-seq data for CRX (*Corbo et al., 2010*), NRL (*Hao et al., 2012*), and OTX2 (anti-OTX2 dataset; *Samuel et al., 2014*). In order to identify putative TF binding sites, we used GEM (*Guo et al., 2012*) with the following parameters: -d Read_Distribution_default.txt –k_min 6 –k_max 15. For NRL, we also specified a seed motif (–seed TCAGCA) because otherwise GEM reported the canonical CRX motif (GGATTA) as the most significant. To calculate the percentage of TF ChIP-seq peaks that overlap ATAC-seq peaks by ≥1 bp (*Figure 3E*; *Figure 3—figure supplement 1C*), we extended the peak centers identified by GEM by 100 bp on each side.

We used SICER_V1.1 (*Zang et al., 2009*) in order to find H3K4me1, H3K4me3, H3K27ac, and H3K27me3 ChIP-seq peaks at FDR = 0.001. We ran SICER with the following parameters: redundancy threshold=1; fragment size=150; W=200, G=200 for H3K4me1, H3K4me3, and H3K27ac; and W=200, G=1000 for H3K27me3. The unions of SICER peaks across replicates for each histone modification are shown in *Supplementary file 8*. In order to assess the similarity of ChIP-seq replicate pairs, we calculated the input-subtracted ChIP-seq signal of each replicate over the union of the two replicates' SICER peaks. We then calculated the Pearson correlation between replicates (shown in *Supplementary file 1*).

## DNaseI-seq

DNaseI-seq data was obtained from the mouse ENCODE project (*Stamatoyannopoulos et al., 2012*). For P1, P7, and P56 C57BL/6J retinas, we used BOWTIE v0.12.7, options -m 1 in order to align reads and identified DNaseI-seq peaks using MACS2 with the same parameters as for ATAC-seq.

## RNA-seq

For RNA-seq data processing, reads from purified WT rods, *rd7* rods, and cones (this study) as well as WT retina and *Nrl KO* retina [data from *Brooks et al. (2011)*] were aligned to the whole genome (TOPHAT v1.4.0) for display in the AnnoJ browser and to the mm10 iGenomes transcriptome annotation. The latter was used for estimating gene expression levels at protein-coding genes using RSEM v1.1.20 (*Li and Dewey, 2011*) calling BOWTIE v0.12.7. Differentially expressed genes at 5% FDR were identified using EBSeq v1.1 (*Leng et al., 2013*). Both TPM values from RSEM and the EBSeq output of differentially expressed genes are shown in *Supplementary file 5*.

We applied more stringent definitions of differential gene expression in order to focus on patterns of chromatin accessibility and DNA methylation at the most highly rod- or cone-specific genes. For several panels (i.e., *Figure 3C*; *Figure 3—figure supplement 1B*; *Figure 6A–B*, *Figure 6C* [left and middle]), we used expressed (TPM ≥30) genes with at least five-fold difference in expression levels between WT retina and *Nrl KO* retina [re-analyzed data from *Brooks et al. (2011)*]. Genes with higher expression in WT retina were called as rod-specific whereas genes with higher expression in *Nrl KO* retina was called as cone-specific. For comparison, we used all genes with TPM ≥30 in both WT retina and *Nrl KO* retina in *Figure 6C*, right. Asterisks in *Figure 6D,E*, *Figure 6—figure supplement 2*, and *Figure 6—figure supplement 3* indicate pairwise comparisons where each individual gene was required to have a posterior probability of differential expression (PPDE) >0.99. In *Figure 5—figure supplement 1A*, we used genes with ≥2-fold expression difference.

## TF motifs

To identify predictive DNA sequences in ATAC-seq peaks, we compared 500 bp regions of the genome between WT retina and *Nrl KO* retina. A sample-specific peak required five-fold fewer reads whereas a shared peak required less than 30% variation. In addition, each region was required to contain ≥15 ATAC-seq reads. We omitted 500 bp regions within 2 kb of the TSS, and each region was positioned to maximize the ATAC-seq signal.

MotifSpec (*Karnik and Beer, 2015*) was run in discriminative mode with these sets in all pairwise combinations, using motif lengths 10, 12 and 15. The most predictive motifs were further analyzed by scoring each motif in each sequence set and generating an ROC curve to assess the ability of the motif to correctly classify the pair of sets. The most predictive motifs were scored against motif databases using TOMTOM (*Gupta et al 2007*) with similarity q-value <0.01. Gkm-SVM (*Ghandi et al., 2014*) was run with standard parameters l=10, k=6, and d=3 on WT-specific, *Nrl KO*-specific, and shared 500 bp sequences and on GC-matched random genomic sequences. The SVM score is $SVM = \sum_i w_i x_i$ where $w_i$ is the weight for each 10-mer, and $x_i$ is the count for each 10-mer in the sequence.

We used the *k*-mer regulatory vocabulary inferred by gkm-SVM trained on chromatin accessibility from ATAC-seq, DNaseI hypersensitive sites, and other enhancer marks in different cell types in order to assess whether the gkm-SVM score could predict retinal enhancer activity in a massively parallel reporter assay (*Shen et al., 2016*). Specifically, we used WT retina ATAC-seq (our study), DNaseI hypersensitive sites (DHS) from 8 week old retina (*Yue et al., 2014*), and enhancer marks in three unrelated cell types: P300-bound enhancers in melanocytes (*Gorkin et al., 2012*), GATA1-bound enhancers in megakaryocytes (*Pimkin et al., 2014*), and DHS regions in lymphoblasts (*Lee et al., 2015*). We trained the gkm-SVM classifier using l = 10 and k = 6 on the top 4000 distal (≥2 kb away from a TSS) WT retina ATAC-seq peaks, the top 10,000 distal retina DHSs, the top 2351 P300-bound melanocyte enhancers, the top 1230 megakaryocyte GATA1-bound enhancers, and the top 22384 GM12878 DHSs versus length, repeat, and GC-matched negative sets of 16000, 10,000, 9404, 4920, and 22384 sequences, respectively. *Shen et al., 2016* included 36,005 constructs, each of which produced at least 10 raw DNA reads in all three biological replicates and RNA

barcodes that could be detected in zero to three replicates. We used the number of replicates in which RNA was detected to discretize the expression level, as we found that this captured most of the expression variance. Then, using the regulatory vocabulary trained by applying gkm-SVM on the various datasets, we scored each full construct and compared the gkm-SVM score to the observed retinal expression level.

## Data access
Data files are available at GEO accession GSE72550 and can also be viewed at http://neomorph. salk.edu/mm_retina/.

## Acknowledgements

We thank Yun-Zheng Le (University of Oklahoma Health Sciences Center), Anand Swaroop (NEI), Donald Zack (Johns Hopkins), and Patsy Nishina (JAX) for their generosity in sharing mice and Hao Zhang at the Johns Hopkins School of Public Health for flow sorting. We thank Jamie Simon for assistance with illustrations. We thank Amir Rattner and Chris Cho for critical reading of the manuscript.

## Additional information

### Competing interests
SRE, JN: Reviewing editor, *eLife* The other authors declare that no competing interests exist.

### Funding

| Funder | Grant reference number | Author |
| --- | --- | --- |
| National Institute of General Medical Sciences | Medical Scientist Training Program (T32) | Alisa Mo |
| National Institutes of Health | Visual Neuroscience Training Program | Alisa Mo |
| Howard Hughes Medical Institute | | Chongyuan Luo<br>Sean R Eddy<br>Joseph R Ecker<br>Jeremy Nathans<br>Fred P Davis<br>Gilbert L Henry<br>Serge Picard |
| National Institute of Neurological Disorders and Stroke | R00NS080911 | Eran A Mukamel |
| National Institutes of Health | R01HG0007348 | Michael A Beer |
| National Institute of Mental Health | 1-U01-MH105985 | Joseph R Ecker |
| Gordon and Betty Moore Foundation | GBMF3034 | Joseph R Ecker |

The funders had no role in study design, data collection and interpretation, or the decision to submit the work for publication.

### Author contributions
AM, Conception and design, Acquisition of data, Analysis and interpretation of data, Drafting or revising the article; CL, FPD, EAM, RL, MAB, Analysis and interpretation of data, Drafting or revising the article; GLH, JRN, MAU, SP, Acquisition of data, Drafting or revising the article; SRE, JRE, JN, Conception and design, Drafting or revising the article

### Author ORCIDs

Ryan Lister, http://orcid.org/0000-0001-6637-7239
Sean R Eddy, http://orcid.org/0000-0001-6676-4706

Joseph R Ecker, http://orcid.org/0000-0001-5799-5895
Jeremy Nathans, http://orcid.org/0000-0001-8106-5460

## Ethics

Animal experimentation: This study was performed in strict accordance with the recommendations in the Guide for the Care and Use of Laboratory Animals of the National Institutes of Health. All animals were handled according to approved institutional animal care and use committee (IACUC) protocol MO13M469 of the Johns Hopkins Medical Institutions.

# Additional files

### Supplementary files

• Supplementary file 1. Characteristics of each sequencing sample

• Supplementary file 2. Hypo-methylated features in each sample. (A-C) UMRs for each methylome sample; (D-F) LMRs for each methylome sample

• Supplementary file 3. Accessible chromatin peaks in each sample (A-J) ATAC-seq peaks for each sample (narrowPeak files from MACS2).

• Supplementary file 4. Differentially methylated regions. (A) Pairwise DSS DMRs between rods and cones, hypo-DMR in rods; (B) Pairwise DSS DMRs between rods and cones, hypo-DMR in cones; (C) Pairwise DSS DMRs between WT rods and rd7 rods, hypo-DMR in WT rods; (D) Pairwise DSS DMRs between WT rods and rd7 rods, hypo-DMR in rd7 rods; (E) Pairwise DSS DMRs between rd7 rods and cones, hypo-DMR in rd7 rods; (F) Pairwise DSS DMRs between rd7 rods and cones, hypo-DMR in cones; (G) DMRs between retina and cortex samples identified using Methylpy. In column D, a 1 denotes the hypo-methylated sample(s), in the following order: neocortical excitatory neurons, PV neurons, and VIP neurons (from Mo et al., 2015), Cones, WT rods, and rd7 rods (this study).

• Supplementary file 5. Gene expression data. (A) RNA abundances (in TPM) for each RNA-seq sample (rods, rd7 rods, and cones from this study; re-analysis of WT retina and Nrl KO retina from Brooks et al., 2011). (B-E) Pairwise lists of differentially expressed genes.

• Supplementary file 6. Summary of regions selected for in vivo electroporation

• Supplementary file 7 Characteristics of DNA methylation valleys

• Supplementary file 8. Rod histone modifications. (A-D) SICER peaks for H3K27ac (A), H3K4me1 (B), H3K4me3 (C), and H3K27me3 (D) enrichment in WT rods.

### Major datasets

The following dataset was generated:

| Author(s) | Year | Dataset title | Dataset URL | Database, license, and accessibility information |
|---|---|---|---|---|
| Mo A, Luo C, Davis FP, Mukamel E, Eddy SR, Ecker JR, Nathans J | 2015 | Epigenomic Landscapes of Retinal Rods and Cones | http://www.ncbi.nlm.nih.gov/geo/query/acc.cgi?acc=GSE72550 | GSE72550 |

The following previously published datasets were used:

| Author(s) | Year | Dataset title | Dataset URL | Database, license, and accessibility information |
|---|---|---|---|---|
| Brooks MJ, Rajasimha HK, Roger JE, Swaroop A | 2011 | Next Generation Sequencing Facilitates Quantitative Analysis of Wild Type and Nrl-/- Retinal Transcriptomes | http://www.ncbi.nlm.nih.gov/geo/query/acc.cgi?acc=GSE33141 | GSE33141 |
| Langmann T, Corbo JC | 2010 | Deciphering the cis-regulatory architecture of mammalian photoreceptors | http://www.ncbi.nlm.nih.gov/geo/query/acc.cgi?acc=GSE20012 | GSE20012 |
| Sandstrom R | 2012 | DNaseI Hypersensitivity by Digital DNaseI from ENCODE/University of Washington | http://www.ncbi.nlm.nih.gov/geo/query/acc.cgi?acc=GSE37074 | GSE37074 |
| Hao H, Kim DS, Klocke B, Johnson KR, Cui K, Gotoh N, Zang C, Gregorski J, Gieser L, Peng W, Fann Y, Seifert M, Zhao K, Swaroop A | 2012 | Hong PLoS-Genet-2012 | http://datashare.nei.nih.gov/nnrlMain.jsp | n/a |
| Lamonerie T, Samuel A, Housset M | 2014 | Otx2 ChIP-seq in the adult mouse Retinal Pigmented Epithelium (RPE) and neural retina | http://www.ncbi.nlm.nih.gov/geo/query/acc.cgi?acc=GSE54084 | GSE54084 |
| Shen Y, Yue F, Ren B | 2012 | A draft map of cis-regulatory sequences in the mouse genome | http://www.ncbi.nlm.nih.gov/geo/query/acc.cgi?acc=GSE29184 | GSE29184 |
| Lister R, Ecker JR | 2013 | Global epigenomic reconfiguration during mammalian brain development | http://www.ncbi.nlm.nih.gov/geo/query/acc.cgi?acc=GSE47966 | GSE47966 |
| Mo A, Mukamel EA, Davis FP, Luo C, Eddy SR, Ecker JR, Nathans J | 2015 | Epigenomic Signatures of Neuronal Diversity in the Mammalian Brain | http://www.ncbi.nlm.nih.gov/geo/query/acc.cgi?acc=GSE63137 | GSE63137 |

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
