## [Decision Letter]

[Editors’ note: this article was originally rejected after discussions between the reviewers, but the authors were invited to resubmit as a Tools and Resources article after an appeal against the decision.]

Thank you for submitting your work entitled "Epigenomic landscapes of retinal rods and cones" for consideration by *eLife*. Your article has been reviewed by three peer reviewers, and the evaluation has been overseen by a Reviewing Editor and a Senior Editor. Our decision has been reached after consultation between the reviewers. Based on these discussions and the individual reviews below, we regret to inform you that your work will not be considered further for publication in *eLife*.

The work provides an excellent dataset for those in the field, and beyond, as well as provides an example for others who wish to carry out a comprehensive and rigorous assessment of the chromatin of a specific cell type. The data also provide a more in depth view of photoreceptor biology, including the interesting findings concerning the differences between rods and cones, with implications regarding the evolution and function of photoreceptors.

However, I am sorry to say that the paper was not judged as reaching a novel enough conclusion for *eLife*. The findings regarding rods and cones were not judged significant enough by two of the reviewers. Due to the negative opinion on this point, I did consult another person who is knowledgeable in the field of retinal biology, to see if I could find additional support. Though this individual was not able to perform a full review, he did look it over for the overall conclusions, and again, the opinion was essentially the same. However, the work is beautiful and for a fan of photoreceptors, very interesting.

*Reviewer #1:* This manuscript details the epigenetic landscape and correlative features of separated retinal rod and cone cells, as distinguished from studies of the whole retina. This appears to be the first study of this kind. The authors perform comprehensive epigenomic profiling separately integrating ATAC-seq, DNA methylation, RNA-seq and histone modifications. A main result is the unusual pattern specific for rods of unmethylated reigons in closed chromatin and without enhancer activity, but with enhancer activity at a prior stage (fetal). This idea is not entirely novel as it has previously been called vestigial enhancers. The computational analyses are standard except for the use of a new Motif tool (MotifSpec) which is to be described elsewhere (though the advantage of the tool relative to what is available is unknown).

The paper is well written and the findings appear solid with conventional computational correlations. The major issue for this reviewer is the cursory attention to function. Figure 4 provides the only functional data which shows that ATAC-seq regions often provide enhancer activity. The extent of enhancer analysis is not up to what can be done to go beyond mere description. In weighing strengths and weaknesses for *eLife* publication here one has to balance the overall quality of the descriptive data with the relative lack of really new insights into epigenetics and function. Frankly I am on the fence as to whether this manuscript meets criteria for *eLife* or its more suitable for a genomics journal. The datasets will be useful to workers in the field but probably not more widely.

[Minor comments not shown.]

*Reviewer #2:*

This research article by Mo et al. continues exploration of cell type-specific epigenomic landscapes, building on their own recently published work in Neuron (NeuroResource: Epigenomic Signatures of Neuronal Diversity in the Mammalian Brain; http://dx.doi.org/10.1016/j.neuron.2015.05.018).

As shown in that earlier publication, combined analysis of DNA methylation and chromatin accessibility is an effective way to characterize cell type-specific regulatory elements. The authors apply this strategy by combining MethylC-Seq and ATAC-Seq analysis to nuclei isolated by FACS sorting or affinity purification (INTACT, a technique they adapted to mammalian cells in their Neuron paper). Unlike that earlier work, this new paper does not introduce a novel, widely applicable technical approach such as INTACT.

The biological observation that the report centers on is that chromatin in rods has hypomethylated regions that correspond to regulatory loci playing important roles in the fetal neuronal development. The authors argue convincingly that these regions are likely shielded from methylation due to the (well-characterized) unusually compact chromatin structure in rod cells. The authors also use comparisons between wildtype and *Nr2e3 KO (rd7*) rods and between *rd7* and *Nrl KO* (with massive shift to cones) retinas to argue that *rd7* rods acquire intermediate epigenetic landscapes between rods and cones.

This is a generally solid descriptive paper which does not happen to yield a new biological insight. It thus appears to fall short of *eLife*'s stated goal of publishing highly significant work. If the editors feel this work belongs in *eLife*, perhaps it might be a better fit to the Tools and resources section. (Cf. a recent publication in that section: "Cell type-specific transcriptomics of hypothalamic energy-sensing neuron responses to weight-loss"; http://elifesciences.org/content/4/e09800).

Specific comments:

H3K27me3 ChIP-Seq: one of the replicates has very low coverage (3.5M deduplicated aligned reads vs. 16.5M in the other experiment; [Supplementary-material SD1-data]). This is likely indicative of a technical failure. At the very least, combining data from these two replicates should be explained; better, the replicate should be redone or removed.

Electroporation analysis of regulatory regions is not particularly convincing. Shouldn't rod- or cone-specific expression lead to a large number of cells with signal (as in Nr2e3), regardless of signal strength? So it is unclear why native GFP was preferred to AP. The scoring is also confusing. Based on Figure 4, it is unclear why signal from Pde6h(-4.6kb) fragment is called "moderate" while *Pde6h*(-0.8kb) is called "weak" (in [Supplementary-material SD6-data]). Even if we fully accept the authors' interpretation, there is no particular reason to think that the regions tested by electroporation are representative of all regions used for motif analysis. It might make more sense to first perform the motif finding, and then use the regions with most characteristic motifs to test experimentally.

Authors note that the INTACT analysis was less cell specific in the retina, because after isolation, nuclei typically formed aggregates (possibly with nuclei from cells of different types), rather than stay as singlets. Since cell specificity of the analysis is a major point, it would be important to understand how much of a distortion is introduced by this aggregation. What is the distribution of nuclei per the singlet, doublet, etc. bins? How often doublets and higher order aggregates contain both rod and cone nuclei (perhaps even rough analysis based on the nuclear appearance might suffice)?

[Minor comments not shown.]

*Reviewer #3:* This is a very novel study which, for the first time, presents a comprehensive analysis of both global as well as specific/local epigenomic states in rods and cones. Although the work is a bit less mechanistic than would be desired, the vast array of data and comprehensive, broad spanning nature of the work makes this a strong manuscript, suitable for publication in *eLife*. The focus on NR2E3 loss is quite interesting and impactful; it would have been great if they could identify additional combinations of genes which (when lost/modified) promote the cone-like state to a greater extent/achieve more complete conversion. While this (the rod/cone biology component) is slightly removed from my field of expertise, based on the work presented and the conclusions derived, I think this paper should be accepted, pending minor revisions.

This is without question the most comprehensive analysis of the epigenomic features of rods and cones, and the first to generate such data in cones, which are far less abundant than rods in the retina. The authors have the difficult task of bringing forward the key, focused points (i.e. the studies on NR2E3) while showing massive amounts of data that were needed to inform more specific mechanisms.

Are there additional mechanistic insights that have been gained since submission of the manuscript with respect to the loss of NR2E3? Or cooperating genes which would further promote a cone-like state?

Technical comments:

Figure 1.

A) Could the authors extend this to show not only a gene expressed by both but other genes expressed in either rods or cones? The general accessibility patterns would be valuable to see as comparison here.

B) Are there ways to indicate the statistical significance on the figure for B and C? The curves appear very close, and hence, some measurement of significance here would be helpful, if possible. Perhaps a bar graph for site distance next to these graphs?

C) In Figure 1, is there anything you can generalize about these specific sites in this figure to substantiate the finding? Or show tracks next to the bar graph to indicate what this looks like on key genes?

Figure 2.

Could the authors simplify Figure 2? Reduce the # tracks? The key point is a bit lost in the number of tracks shown here. Best to separate between this figure and Figure 2—figure supplement 1.

Figure 4.

To complement this, are there quantitative summaries that could be displayed in the figure, i.e.% GFP expression for the key comparisons? Also, to save space/redundancy, perhaps the authors could show just 2 of the 'near cone genes' and place the rest of these in supplemental? I think the point is well illustrated and quite clear.

If possible, as part of Figure 7 model/summary figure would improve the paper and allow more citations via highlighting the model. I think this would be an important addition.

---

## [Author Response]

We are gratified by the generally positive comments from the three reviewers, and we are therefore surprised that the decision was not more favorable. This is the first report of a genome-wide comparison between and integrated analysis of the methylomes and the accessible chromatin landscapes in rods and cones. It is also only the second report of such a comparison between purified neuronal cell types, the first report being our Neuron paper earlier this year describing that comparison for subsets of cortical neurons. Additionally, the present manuscript pushes the purification, methylome sequencing, and chromatin accessibility technologies to a new level by targeting a cell type (cones) that is rarer than any cell type to which these methods have yet been applied. The high quality datasets that we have generated, the extensive computational analyses that we have performed, the whole genome nature of the insights derived from those analyses, and the surprising differences that we have uncovered between rod and cone epigenomes, all set the present study apart from everything else in the scientific literature on sensory receptor cells.

Given the central importance of rods and cones in vision research and in clinical ophthalmology, as well as their central place in the larger field of sensory neuroscience, we predict that this work will be of broad and enduring interest. The editor's letter states "The work is beautiful and for a fan of photoreceptors, very interesting." We agree, and we would add that the study of vertebrate photoreceptors is not a niche field. A recent PubMed search shows >7000 papers with the search term 'cone photoreceptor' and >10,000 papers with the search term 'rod photoreceptor'.

Reviewer #2 suggested that this manuscript might be a better fit as a Tools and Resources article instead of a regular article. We think that this is an excellent suggestion, since the "Tools and Resources" paragraph of the eLife "Information for Authors" section states: "For example, we welcome the submission of.. . genomic or other datasets..." If you agree that this is a reasonable path forward, we would be happy to submit a revised version that addresses the various technical points raised by the reviewers and that adjusts the text to the Tools and Resources venue.

Responses to the individual reviews follow.

Reviewer #1:

[…] The paper is well written and the findings appear solid with conventional computational correlations. The major issue for this reviewer is the cursory attention to function. Figure 4 provides the only functional data which shows that ATAC-seq regions often provide enhancer activity. The extent of enhancer analysis is not up to what can be done to go beyond mere description. In weighing strengths and weaknesses for eLife publication here one has to balance the overall quality of the descriptive data with the relative lack of really new insights into epigenetics and function. Frankly I am on the fence as to whether this manuscript meets criteria for eLife or its more suitable for a genomics journal. The datasets will be useful to workers in the field but probably not more widely.

As described below, the revised manuscript includes additional analysis validating the functional role of motifs predicted by our computational analysis (gkm-SVM). We believe this new analysis, taking advantage of a large-scale dataset published while our paper was under review, greatly bolsters the functional relevance of our results and should address the reviewer’s concern.

Reviewer #2:

*[…] This is a generally solid descriptive paper which does not happen to yield a new biological insight. It thus appears to fall short of eLife's stated goal of publishing highly significant work. If the editors feel this work belongs in eLife, perhaps it might be a better fit to the Tools and resources section. (Cf. a recent publication in that section: "Cell type-specific transcriptomics of hypothalamic energy-sensing neuron responses to weight-loss"; http://elifesciences.org/content/4/e09800).* We respectfully disagree with the reviewer’s characterization of our study. We integrate an enormous amount of new epigenomic and transcriptomic data with extensive quantitative analyses that offer the first high-resolution (e.g., single base-pair DNA methylation data), genome-wide, and cell type-specific window into the organization of any mammalian sensory neuron. Our analysis yields multiple new biological insights. For example:

1) Despite the close developmental and functional relationship between retinal rods and cones, we show that there is a *profound difference in the organization of their DNA methylation landscapes*. In particular, rod photoreceptors, but not cones, show a substantial discrepancy between regions of low DNA methylation and of accessible chromatin. This is the first observation of such a widespread dissociation between chromatin accessibility and DNA methylation in a mammalian cell type.

2) Sophisticated computational approaches are used to identify DNA sequence features of accessible chromatin in rods and cones, including the discovery of a novel motif in rod-specific accessible chromatin.

3) In the revised manuscript, we show that scores obtained from training a machine learning algorithm on retinal accessible chromatin peaks are strongly predictive of functional enhancer activity.

4) Extending beyond the retina, we find that, despite the anatomical difference, DNA methylation patterns in the fetal cerebral cortex are more similar to mature retinal photoreceptors than to mature cortical neurons.

*Specific comments: H3K27me3 ChIP-Seq: one of the replicates has very low coverage (3.5M deduplicated aligned reads vs. 16.5M in the other experiment; [Supplementary-material SD1-data]). This is likely indicative of a technical failure. At the very least, combining data from these two replicates should be explained; better, the replicate should be redone or removed.*

We agree that 3.5M reads are low, but our assessment of this data shows extremely good agreement with the dataset that has 16.5M reads. We have now calculated the correlation of ChIP-seq signals between replicate pairs and find high correlations (Pearson r = 0.96 – 1.0) between all pairs, including H3K27me3. For this assessment, we evaluated the ChIP-seq signals over the union of peaks called in each individual replicate (for example, input-subtracted H3K27me3 replicate 1 and replicate 2 signals over the union of their SICER peaks; r = 0.97). Therefore, we believe that combining data from the two replicates is justified. We have added the correlations to [Supplementary-material SD1-data].

Electroporation analysis of regulatory regions is not particularly convincing. Shouldn't rod- or cone-specific expression lead to a large number of cells with signal (as in Nr2e3), regardless of signal strength? So it is unclear why native GFP was preferred to AP. The scoring is also confusing. Based on Figure 4, it is unclear why signal from Pde6h(-4.6kb) fragment is called "moderate" while Pde6h(-0.8kb) is called "weak" (in [Supplementary-material SD6-data]). Even if we fully accept the authors' interpretation, there is no particular reason to think that the regions tested by electroporation are representative of all regions used for motif analysis. It might make more sense to first perform the motif finding, and then use the regions with most characteristic motifs to test experimentally.

In this experiment, we consistently found that weak regulatory sites gave fewer cells with detectable native GFP signal. This observation is most likely due to the variation in the number of plasmids taken up by each cell. We hypothesize that, for plasmids containing a weak enhancer, only those few cells that have taken up a large number of plasmids are detectably GFP+. On the other hand, strong enhancers give an appreciable GFP signal in many cells, even those that have taken up a small number of plasmids. Supporting this hypothesis, we typically observe a wide range of signal intensities in both the GFP and TdT channels, even when comparing adjacent cells in the same retina.

Either GFP or AP staining could have been used. We chose to use GFP because it offered greater precision in quantifying the number of labeled cells than AP staining. Furthermore, native GFP and TdT signals are linearly related to protein levels, whereas immunostaining or an enzymatic reporter would introduce a non-linearity. We have now simplified the categories into “strong,” “weak,” and “none.” We have further quantified the % of electroporated (TdT+) cells that are GFP+ for each construct and added it to the figure and to [Supplementary-material SD6-data].

*Authors note that the INTACT analysis was less cell specific in the retina, because after isolation, nuclei typically formed aggregates (possibly with nuclei from cells of different types), rather than stay as singlets. Since cell specificity of the analysis is a major point, it would be important to understand how much of a distortion is introduced by this aggregation. What is the distribution of nuclei per the singlet, doublet, etc. bins? How often doublets and higher order aggregates contain both rod and cone nuclei (perhaps even rough analysis based on the nuclear appearance might suffice)?*

We would first like to clarify that GFP+ nuclei from a Cre-recombined R26-CAG-LSL-Sun1-sfGFP-myc mouse can either be isolated by using affinity purification (as described in Mo et al., 2015) or by using FACS. Because the GFP tag is tethered to the nuclear membrane, the fluorescence remains associated with the nuclei throughout the FACS procedure. We isolated cone nuclei using FACS because FACS can differentiate between singlets and higher-order aggregates (Figure 1—figure supplement 1). Therefore, our purified cone nuclei are free of nuclear aggregates.

We note that there is a practical advantage of affinity purification over FACS. In our experience, affinity-purified nuclei are easier to manipulate than FACS-sorted nuclei for subsequent ATAC-seq reactions.

Our FACS profile of the distribution of singlet and higher-order nuclei after the initial homogenization step (performed identically for both the FACS- and INTACT-purified nuclei, with the latter samples further processed through a 20 μm filter; Figure 1—figure supplement 1) shows that the large majority of nuclei are singlets. Moreover, most of the doublets and higher-order aggregates are likely to contain only rods because the majority of the retina is composed of rods. Taken together, this implies that affinity purification of rods will be highly specific. We have now calculated and added to the text that, based on fluorescence microscopy of purified nuclei (>400 nuclei counted per experiment), INTACT purification of rod photoreceptors was 97.5% specific (96.7 – 98.2%; n = 5). Essentially 100% of the non-rod nuclei arose from higher-order aggregates, rather than from contaminating singlets.

Reviewer #3:

*This is a very novel study which, for the first time, presents a comprehensive analysis of both global as well as specific/local epigenomic states in rods and cones. Although the work is a bit less mechanistic than would be desired, the vast array of data and comprehensive, broad spanning nature of the work makes this a strong manuscript, suitable for publication in eLife. The focus on NR2E3 loss is quite interesting and impactful; it would have been great if they could identify additional combinations of genes which (when lost/modified) promote the cone-like state to a greater extent/achieve more complete conversion. While this (the rod/cone biology component) is slightly removed from my field of expertise, based on the work presented and the conclusions derived, I think this paper should be accepted, pending minor revisions.*

We thank the reviewer for this appreciation of our study’s novelty and impact. Indeed, we went to great lengths to comprehensively analyze both global and local genomic regulatory features of the rod and cone datasets we generated, including by applying advanced computational tools such as gkm-SVM.

*This is without question the most comprehensive analysis of the epigenomic features of rods and cones, and the first to generate such data in cones, which are far less abundant than rods in the retina. The authors have the difficult task of bringing forward the key, focused points (i.e. the studies on NR2E3) while showing massive amounts of data that were needed to inform more specific mechanisms.*

*Are there additional mechanistic insights that have been gained since submission of the manuscript with respect to the loss of NR2E3? Or cooperating genes which would further promote a cone-like state?*

To add additional mechanistic information into the remodeling process, we extended our *k*-mer analysis to identify motifs that could distinguish *rd7*-specific ATAC-seq peaks from random regions, as well as shared *rd7* and WT retina ATAC-seq peaks from random regions (Figure 10). We find that the sets of enriched sequences are similar between the two comparisons, as well as to our previous comparisons using WT and *Nrl KO* retina, including large SVM weights for *k*-mers matching the CRX binding site (GATTA). In contrast, the sets of depleted sequences are substantially different (albeit, with lower weights).

*Technical comments:*

Figure 1.

*A) Could the authors extend this to show not only a gene expressed by both but other genes expressed in either rods or cones? The general accessibility patterns would be valuable to see as comparison here.*

We agree that visual examples of different types of genes are helpful. However, our main goal for Figure 1 is to present an example of our various types of sequencing datasets. In later figures (Figure 3, Figure 6, and their supplemental figures), we show genes with different expression patterns.

*B) Are there ways to indicate the statistical significance on the figure for B and C? The curves appear very close, and hence, some measurement of significance here would be helpful, if possible. Perhaps a bar graph for site distance next to these graphs?*

We apologize for the lack of clarity in this figure. Figure 1 shows that rods and cones both have lower mean DNA methylation levels at their respective ATAC-seq peaks. Figure 1 shows that rods and cones both have higher ATAC-seq levels at their respective un- and low-methylated regions. Rather than comparing the green and purple curves to each other, these curves should be compared to the gray curves that depict CG methylation (1B) and ATAC-seq signal (1C) at randomly selected genomic regions (please note that what might appear as a single gray curve is actually multiple curves from random sampling of matched genomic regions). There is a striking difference between either the green or purple curve and the gray curves. Therefore, we believe that showing the magnitude of effect is more helpful than the statistical significance, given that the significance would be predominantly driven by the large size of the datasets.

C) In Figure 1, is there anything you can generalize about these specific sites in this figure to substantiate the finding? Or show tracks next to the bar graph to indicate what this looks like on key genes?

We present our in-depth integration of chromatin accessibility with DNA methylation using hypo-DMRs (Figure 2), rather than using UMRs and LMRs (Figure 1), because there is a greater discrepancy between hypo-DMRs and ATAC-seq peaks (Figure 2) than between UMRs/LMRs and ATAC-seq peaks (Figure 1). Because rod hypo-DMRs are largely a subset of rod UMRs and LMRs (74% of rod hypo-DMRs are included in the set of rod UMRs+LMRs), this discrepancy suggests that vestigial enhancer locations are uniquely hypo-methylated in rods. We expect that analysis using UMRs and LMRs would yield similar, but less dramatic, results.

Figure 2.

Could the authors simplify Figure 2? Reduce the # tracks? The key point is a bit lost in the number of tracks shown here. Best to separate between this figure and Figure 2—figure supplement 1.

We experimented with splitting up the tracks between this figure and the supplement. Ultimately, we decided to leave Figure 2 as is, because in our view, all the tracks are important to the figure.

Figure 4.

*To complement this, are there quantitative summaries that could be displayed in the figure, i.e.% GFP expression for the key comparisons? Also, to save space/redundancy, perhaps the authors could show just 2 of the 'near cone genes' and place the rest of these in supplemental? I think the point is well illustrated and quite clear.*

Thank you for the suggestion. We have now quantified the % of electroporated (TdT+) cells that are GFP+ for each construct and added it to the figure, as well as to [Supplementary-material SD6-data]. As you suggested, we now show two of the ‘near cone genes’ in the main figure and have placed the rest in Figure 4—figure supplement 1.

If possible, as part of Figure 7 model/summary figure would improve the paper and allow more citations via highlighting the model. I think this would be an important addition.

Thank you. We have added a model/summary figure. To highlight the significance of the model, we have placed it as a separate figure (Figure 9).

Author response image 1.**DOI:**
http://dx.doi.org/10.7554/eLife.11613.036